# The Label Complexity of Active Learning from Observational Data

**Songbai Yan**
University of California San Diego
yansongbai@eng.ucsd.edu

**Kamalika Chaudhuri**
University of California San Diego
kamalika@cs.ucsd.edu

**Tara Javidi**
University of California San Diego
tjavidi@eng.ucsd.edu

## Abstract

Counterfactual learning from observational data involves learning a classifier on an entire population based on data that is observed conditioned on a selection policy. This work considers this problem in an active setting, where the learner additionally has access to unlabeled examples and can choose to get a subset of these labeled by an oracle.

Prior work on this problem uses disagreement-based active learning, along with an importance weighted loss estimator to account for counterfactuals, which leads to a high label complexity. We show how to instead incorporate a more efficient counterfactual risk minimizer into the active learning algorithm. This requires us to modify both the counterfactual risk to make it amenable to active learning, as well as the active learning process to make it amenable to the risk. We provably demonstrate that the result of this is an algorithm which is statistically consistent as well as more label-efficient than prior work.

## 1 Introduction

Counterfactual learning from observational data is an emerging problem that arises naturally in many applications. In this problem, the learner is given observational data – a set of examples selected according to some policy along with their labels – as well as access to the policy that selects the examples, and the goal is to construct a classifier with high performance on an entire population, not just the observational data distribution. An example is learning to predict if a treatment will be effective based on features of a patient. Here, we have some observational data on how the treatment works for patients that were assigned to it, but if the treatment is given only to a certain category of patients, then the data is not reflective of the population. Thus the main challenge in counterfactual learning is how to counteract the effect of the observation policy and build a classifier that applies more widely.

This work considers counterfactual learning in the active setting, which has received very recent attention in a few different contexts [25, 21, 3]. In addition to observational data, the learner has an online stream of unlabeled examples drawn from the underlying population distribution, and the ability to selectively label a subset of these in an interactive manner. The learner's goal is to again build a classifier while using as few label queries as possible. The advantage of the active over the passive is its potential for more label-efficient solutions; the question however is how to do this algorithmically.

Prior work in this problem has looked at both probabilistic inference [21, 3] as well as a standard classification [25], which is the setting of our work. [25] uses a modified version of disagreement-

based active learning [7, 9, 4, 11], along with an importance weighted empirical risk to account for the population. However, a problem with this approach is that the importance weighted risk estimator can have extremely high variance when the importance weights – that reflect the inverse of how frequently an instance in the population is selected by the policy – are high; this may happen if, for example, certain patients are rarely given the treatment. This high variance in turn results in high label requirement for the learner.

The problem of high variance in the loss estimator is addressed in the passive case by minimizing a form of counterfactual risk [22] – an importance weighted loss that combines a variance regularizer and importance weight clipping or truncation to achieve low generalization error. A plausible solution is to use this risk for active learning as well. However, this cannot be readily achieved for two reasons. The first is that the variance regularizer itself is a function of the entire dataset, and is therefore challenging to use in interactive learning where data arrives sequentially. The second reason is that the minimizer of the (expected) counterfactual risk depends on $n$, the data size, which again is inconvenient for learning in an interactive manner.

In this work, we address both challenges. To address the first, we use, instead of a variance regularizer, a novel regularizer based on the second moment; the advantage is that it decomposes across multiple segments of the data set as which makes it amenable for active learning. We provide generalization bounds for this modified counterfactual risk minimizer, and show that it has almost the same performance as counterfactual risk minimization with a variance regularizer [22]. The second challenge arises because disagreement-based active learning ensures statistical consistency by maintaining a set of plausible minimizers of the expected risk. This is problematic when the minimizer of the expected risk itself changes between iterations as in the case with our modified regularizer. We address this challenge by introducing a novel variant of disagreement-based active learning which is always guaranteed to maintain the population error minimizer in its plausible set.

Additionally, to improve sample efficiency, we then propose a third novel component – a new sampling algorithm for correcting sample selection bias that selectively queries labels of those examples which are underrepresented in the observational data. Combining these three components gives us a new algorithm. We prove this newly proposed algorithm is statistically consistent – in the sense that it converges to the true minimizer of the population risk given enough data. We also analyze its label complexity, show it is better than prior work [25], and demonstrate the contribution of each component of the algorithm to the label complexity bound.

## 2   Related Work

We consider learning with logged observational data where the logging policy that selects the samples to be observed is known to the learner. The standard approach is importance sampling to derive an unbiased loss estimator [19], but this is known to suffer from high variance. One common approach for reducing variance is to clip or truncate the importance weights [6, 22], and we provide a new principled method for choosing the clipping threshold with theoretical guarantees. Another approach is to add a regularizer based on empirical variance to the loss function to favor models with low loss variance [17, 22, 18]. Our second moment regularizer achieves a similar effect, but has the advantage of being applicable to active learning with theoretical guarantees.

In this work, in addition to logged observational data, we allow the learner to actively acquire additional labeled examples. The closest to our work is [25], the only known work in the same setting. [25] and our work both use disagreement-based active learning (DBAL) framework [7, 9, 4, 11] and multiple importance sampling [24] for combining actively acquired examples with logged observational data. [25] uses an importance weighted loss estimator which leads to high variance and hence high sample complexity. In our work, we incorporate a more efficient variance-controlled importance sampling into active learning and show that it leads to a better label complexity.

[3] and [21] consider active learning for predicting individual treatment effect which is similar to our task. They take a Bayesian approach which does not need to know the logging policy, but assumes the true model is from a known distribution family. Additionally, they do not provide label complexity bounds. A related line of research considers active learning for domain adaptation, and their methods are mostly based on heuristics [20, 27], utilizing a clustering structure [14], or non-parametric methods [15]. In other related settings, [26] considers warm-starting contextual

bandits targeting at minimizing the cumulative regret instead of the final prediction error; [16] studies active learning with bandit feedback without any logged observational data.

## 3 Problem Setup

We are given an instance space $\mathcal{X}$, a label space $\mathcal{Y} = \{-1, +1\}$, and a hypothesis class $\mathcal{H} \subset \mathcal{Y}^{\mathcal{X}}$. Let $D$ be an underlying data distribution over $\mathcal{X} \times \mathcal{Y}$. For simplicity, we assume $\mathcal{H}$ is a finite set, but our results can be generalized to VC-classes by standard arguments [23, 18].

In the passive setting for learning with observational data, the learner has access to a logged observational dataset generated from the following process. First, $m$ examples $\{(X_t, Y_t)\}_{t=1}^{m}$ are drawn i.i.d. from $D$. Then a logging policy $Q_0 : \mathcal{X} \to [0, 1]$ that describes the probability of observing the label is applied. In particular, for each example $(X_t, Y_t)$ $(1 \leq t \leq m)$, an independent Bernoulli random variable $Z_t$ with expectation $Q_0(X_t)$ is drawn, and then the label $Y_t$ is revealed to the learner if $Z_t = 1$[1]. We call $T_0 = \{(X_t, Y_t, Z_t)\}_{t=1}^{m}$ the logged dataset. We assume the learner knows the logging policy $Q_0$, and only observes instances $\{X_t\}_{t=1}^{m}$, indicators $\{Z_t\}_{t=1}^{m}$, and revealed labels $\{Y_t \mid Z_t = 1\}_{t=1}^{m}$.

In the active learning setting, in addition to the logged dataset, the learner has access to a stream of online data. In particular, there is a stream of additional $n$ examples $\{(X_t, Y_t)\}_{t=m+1}^{m+n}$ drawn i.i.d. from distribution $D$. At time $t$ $(m < t \leq m + n)$, the learner applies a query policy to compute an indicator $Z_t \in \{0, 1\}$, and then the label $Y_t$ is revealed if $Z_t = 1$. The computation of $Z_t$ may in general be randomized, and is based on the observed logged data $T_0$, previously observed instances $\{X_i\}_{i=m+1}^{t}$, decisions $\{Z_i\}_{i=m+1}^{t-1}$, and observed labels $\{Y_i \mid Z_i = 1\}_{i=m+1}^{t-1}$.

We focus on the active learning setting, and the goal of the learner is to learn a classifier $h \in \mathcal{H}$ from observed logged data and online data. Fixing $D, Q_0, m, n$, the performance is measured by: (1) the error rate $l(h) := \Pr_D(h(X) \neq Y)$ of the output classifier, and (2) the number of label queries on the online data. Note that the error rate is over the entire population $D$ instead of conditioned on the logging policy, and that we assume the labels of the logged data $T_0$ come at no cost. In this work, we are interested in the situation where $n$, the size of the online stream, is smaller than $m$.

**Notation** Unless otherwise specified, all probabilities and expectations are over the draw of all random variables $\{(X_t, Y_t, Z_t)\}_{t=1}^{m+n}$. Define $q_0 = \inf_x Q_0(x)$. Define the optimal classifier $h^\star = \arg\min_{h \in \mathcal{H}} l(h)$, $\nu = l(h^\star)$. For any $r > 0, h \in \mathcal{H}$, define the $r-$ball around $h$ as $B(h, r) = \{h' \in \mathcal{H} : \Pr(h(X) \neq h'(X)) \leq r\}$. For any $C \subseteq \mathcal{H}$, define the disagreement region $\mathrm{DIS}(C) = \{x \in \mathcal{X} : \exists h_1 \neq h_2 \in C, h_1(X) \neq h_2(X)\}$.

Due to space limit, all proofs are postponed to Appendix.

## 4 Variance-Controlled Importance Sampling for Passive Learning with Observational Data

In the passive setting, the standard method to overcome sample selection bias is to optimize the importance weighted (IW) loss $l(h, T_0) = \frac{1}{m} \sum_t \frac{\mathbb{1}\{h(X_t) \neq Y_t\} Z_t}{Q_0(X_t)}$. This loss is an unbiased estimator of the population error $\Pr(h(X) \neq Y)$, but its variance $\frac{1}{m} \mathbb{E}(\frac{\mathbb{1}\{h(X) \neq Y\} Z}{Q_0(X)} - l(h))^2$ can be high, leading to poor solutions. Previous work addresses this issue by adding a variance regularizer [17, 22, 18] and clipping/truncating the importance weight [6, 22]. However, the variance regularizer is challenging to use in interactive learning when data arrives sequentially, and it is unclear how the clipping/truncating threshold should be chosen to yield good theoretical guarantees.

In this paper, as an alternative to the variance regularizer, we propose a novel second moment regularizer which achieves a similar error bound to the variance regularizer [18]; and this motivates a principled choice of the clipping threshold.

## 4.1 Second-Moment-Regularized Empirical Risk Minimization

Intuitively, between two classifiers with similarly small training loss $l(h, T_0)$, the one with lower variance should be preferred, since its population error $l(h)$ would be small with a higher probability than the one with higher variance. Existing work encourages low variance by regularizing the loss with the estimated variance $\hat{\text{Var}}(h, T_0) = \frac{1}{m}\sum_i (\frac{\mathbb{1}\{h(X_i)\neq Y_i\}Z_i}{Q_0(X_i)})^2 - l(h, T_0)^2$. Here, we propose to regularize with the estimated second moment $\hat{V}(h, T_0) = \frac{1}{m}\sum_i (\frac{\mathbb{1}\{h(X_i)\neq Y_i\}Z_i}{Q_0(X_i)})^2$, an upper bound of $\hat{\text{Var}}(h, T_0)$. We have the following generalization error bound for regularized ERM.

**Theorem 1.** *Let* $\hat{h} = \arg\min_{h\in\mathcal{H}} l(h, T_0) + \sqrt{\frac{4\log\frac{|\mathcal{H}|}{\delta}}{m}\hat{V}(h, T_0)}$. *For any* $\delta > 0$, *then with probability at least* $1 - \delta$, $l(\hat{h}) - l(h^\star) \leq \frac{28\log\frac{|\mathcal{H}|}{\delta}}{3mq_0} + \sqrt{\frac{4\log\frac{|\mathcal{H}|}{\delta}}{m}\mathbb{E}\frac{\mathbb{1}\{h^\star(X)\neq Y\}}{Q_0(X)}} + \frac{\sqrt{4\log\frac{|\mathcal{H}|}{\delta}}}{m^{\frac{3}{2}}q_0^2}$.

Theorem 1 shows an error rate similar to the one for the variance regularizer [18]. However, the advantage of using the second moment is the decomposability: $\hat{V}(h, S_1 \cup S_2) = \frac{|S_1|}{|S_1|+|S_2|}\hat{V}(h, S_1) + \frac{|S_2|}{|S_1|+|S_2|}\hat{V}(h, S_2)$. This makes it easier to analyze for active learning that we will discuss later.

Recall for the unregularized importance sampling loss minimizer $\hat{h}_{\text{IW}} = \arg\min_{h\in\mathcal{H}} l(h, T_0)$, the error bound is $\tilde{O}(\frac{\log|\mathcal{H}|}{mq_0} + \sqrt{\frac{\log|\mathcal{H}|}{m}\min(\frac{l(h^\star)}{q_0}, \mathbb{E}\frac{1}{Q_0(X)})})$ [8, 25]. In Theorem 1, the extra $\frac{1}{m^{\frac{3}{2}}q_0^2}$ term is due to the deviation of $\sqrt{\hat{V}(h, T_0)}$ around $\sqrt{\mathbb{E}\frac{\mathbb{1}\{h(X)\neq Y\}}{Q_0(X)}}$, and is negligible when $m$ is large. In this case, learning with a second moment regularizer gives a better generalization bound.

This improvement in generalization error is due to the regularizer instead of tighter analysis. Similar to [17, 18], we show in Theorem 2 that for some distributions, the error bound in Theorem 1 cannot be achieved by any algorithm that simply optimizes the unregularized empirical loss.

**Theorem 2.** *For any* $0 < \nu < \frac{1}{3}$, $m \geq \frac{49}{\nu^2}$, *there is a sample space* $\mathcal{X} \times \mathcal{Y}$, *a hypothesis class* $\mathcal{H}$, *a distribution* $D$, *and a logging policy* $Q_0$ *such that* $\frac{\nu}{q_0} > \mathbb{E}\frac{\mathbb{1}\{h^\star(X)\neq Y\}}{Q_0(X)}$, *and that with probability at least* $\frac{1}{100}$ *over the draw of* $S = \{(X_t, Y_t, Z_t)\}_{t=1}^m$, *if* $\hat{h} = \arg\min_{h\in\mathcal{H}} l(h, S)$, *then* $l(\hat{h}) \geq l(h^\star) + \frac{1}{mq_0} + \sqrt{\frac{\nu}{mq_0}}$.

## 4.2 Clipped Importance Sampling

The variance and hence the error bound for second-moment regularized ERM can still be high if $\frac{1}{Q_0(x)}$ is large. This $\frac{1}{Q_0(X)}$ factor arises inevitably to guarantee the importance weighted estimator is unbiased. Existing work alleviates the variance issue at the cost of some bias by clipping or truncating the importance weight. In this paper, we focus on clipping, where the loss estimator becomes $l(h; T_0, M) := \frac{1}{m}\sum_{i=1}^m \frac{\mathbb{1}\{h(X_i)\neq Y_i\}Z_i}{Q_0(X_i)}\mathbb{1}[\frac{1}{Q_0(X_i)} \leq M]$. This estimator is no longer unbiased, but as the weight is clipped at $M$, so is the variance. Although studied previously [6, 22], to the best of our knowledge, it remains unclear how the clipping threshold $M$ can be chosen in a principled way.

We propose to choose $M_0 = \inf\{M' \geq 1 \mid \frac{2M'\log\frac{|\mathcal{H}|}{\delta}}{m} \geq \Pr_X(\frac{1}{Q_0(X)} > M')\}$. This choice minimizes an error bound for the clipped second-moment regularized ERM and we formally show this in Appendix E. Example 30 in Appendix E shows this clipping threshold avoids outputting suboptimal classifiers. The choice of $M_0$ implies that the clipping threshold should be larger as the sample size $m$ increases, which confirms the intuition that with a larger sample size the variance becomes less of an issue than the bias. We have the following generalization error bound.

**Theorem 3.** *Let* $\hat{h} = \arg\min_{h\in\mathcal{H}} l(h; T_0, M_0) + \sqrt{\frac{4\log\frac{|\mathcal{H}|}{\delta}}{m}\hat{V}(h; T_0, M_0)}$. *For any* $\delta > 0$, *with probability at least* $1 - \delta$,

$$l(\hat{h}) - l(h^\star) \leq \frac{34\log\frac{|\mathcal{H}|}{\delta}}{3m}M_0 + \frac{\sqrt{4\log\frac{|\mathcal{H}|}{\delta}}}{m^{\frac{3}{2}}}M_0^2 + \sqrt{\frac{4\log\frac{|\mathcal{H}|}{\delta}}{m}\mathbb{E}\frac{\mathbb{1}\{h^\star(X)\neq Y\}}{Q_0(X)}\mathbb{1}[\frac{1}{Q_0(X)} \leq M_0]}.$$

We always have $M_0 \leq \frac{1}{q_0}$ as $\Pr_X \left( \frac{1}{Q_0(X)} > \frac{1}{q_0} \right) = 0$. Thus, this error bound is always no worse than that without clipping asymptotically.

# 5 Active Learning with Observational Data

Next, we consider active learning where in addition to a logged observational dataset the learner has access to a stream of unlabeled samples from which it can actively query for labels. The main challenges are how to control the variance due to the observational data with active learning, and how to leverage the logged observational data to reduce the number of label queries beyond simply using them for warm-start.

To address these challenges, we first propose a nontrivial change to the Disagreement-Based Active Learning (DBAL) so that the variance-controlled importance sampling objective can be incorporated. This modified algorithm also works in a general cost-sensitive active learning setting which we believe is of independent interest. Second, we show how to combine logged observational data with active learning through multiple importance sampling (MIS). Finally, we propose a novel sample selection bias correction technique to query regions under-explored in the observational data more frequently. We provide theoretical analysis demonstrating that the proposed method gives better label complexity guarantees than previous work [25] and alternative methods.

**Key Technique 1: Disagreement-Based Active Learning with Variance-Controlled Importance Sampling**

The DBAL framework is a widely-used general framework for active learning [7, 9, 4, 11]. This framework iteratively maintains a candidate set $C_t$ to be a confidence set for the optimal classifier. A disagreement region $D_t$ is then defined accordingly to be the set of instances on which there are two classifiers in $C_t$ that predict labels differently. At each iteration, it draws a set of unlabeled instances. The labels for instances falling inside the disagreement region are queried; otherwise, the labels are inferred according to the unanimous prediction of the candidate set. These instances with inferred or queried labels are then used to shrink the candidate set.

The classical DBAL framework only considers the unregularized 0-1 loss. As discussed in the previous section, with observational data, unregularized loss leads to suboptimal label complexity. However, directly adding a regularizer breaks the statistical consistency of DBAL, since the proof of its consistency is contingent on two properties: (1) the minimizer of the population loss $l(h)$ stays in all candidate sets with high probability; (2) the loss difference $l(h_1, S) - l(h_2, S)$ for any $h_1, h_2 \in C_t$ does not change no matter how examples outside the disagreement region $D_t$ are labeled.

Unfortunately, if we add a variance based regularizer (either estimated variance or second moment), the objective function $l(h, S) + \sqrt{\frac{\lambda}{n} \hat{V}(h, S)}$ has to change as the sample size $n$ increases, and so does the optimal classifier w.r.t. regularized population loss $\tilde{h}_n = \arg\min l(h) + \sqrt{\frac{\lambda}{n} V(h)}$. Consequently, $\tilde{h}_n$ may not stay in all candidate sets. Besides, the difference of the regularized loss $l(h_1, S) + \sqrt{\frac{\lambda}{n} \hat{V}(h_1, S)} - (l(h_2, S) + \sqrt{\frac{\lambda}{n} \hat{V}(h_2, S)})$ changes if labels of examples outside the disagreement region $D_t$ are modified, breaking the second property.

To resolve the consistency issues, we first carefully choose the definition of the candidate set and guarantee the optimal classifier w.r.t. the prediction error $h^\star = \arg\min l(h)$, instead of the regularized loss $\tilde{h}_n$, stays in candidate sets with high probability. Moreover, instead of the plain variance regularizer, we apply the second moment regularizer and exploit its decomposability property to bound the difference of the regularized loss for ensuring consistency.

**Key Technique 2: Multiple Importance Sampling**

MIS addresses how to combine logged observational data with actively collected data for training classifiers [2, 25]. To illustrate this, for simplicity, we assume a fixed query policy $Q_1$ is used for active learning. To make use of both $T_0 = \{(X_i, Y_i, Z_i)\}_{i=1}^{m}$ collected by $Q_0$ and $T_1 = \{(X_i, Y_i, Z_i)\}_{i=m+1}^{m+n}$ collected by $Q_1$, one could optimize the unbiased importance weighted error estimator $l_{\text{IS}}(h, T_0 \cup T_1) = \sum_{i=1}^{m} \frac{\mathbb{1}\{h(X_i) \neq Y_i\} Z_i}{(m+n) Q_0(X_i)} + \sum_{i=m+1}^{m+n} \frac{\mathbb{1}\{h(X_i) \neq Y_i\} Z_i}{(m+n) Q_1(X_i)}$ which can have high

variance and lead to poor generalization error. Here, we apply the MIS estimator $l_{\text{MIS}}(h, T_0 \cup T_1) := \sum_{i=1}^{m+n} \frac{\mathbb{1}\{h(X_i) \neq Y_i\} Z_i}{m Q_0(X_i) + n Q_1(X_i)}$ which effectively treats the data $T_0 \cup T_1$ as drawn from a mixture policy $\frac{m Q_0 + n Q_1}{m+n}$. $l_{\text{MIS}}$ is also unbiased, but has lower variance than $l_{\text{IS}}$ and thus gives better error bounds.

**Key Technique 3: Active Sample Selection Bias Correction**

Another advantage to consider active learning is that the learner can apply a strategy to correct the sample selection bias, which improves label efficiency further. This strategy is inspired from the following intuition: due to sample selection bias caused by the logging policy, labels for some regions of the sample space may be less likely to be observed in the logged data, thus increasing the uncertainty in these regions. To counter this effect, during active learning, the learner should query more labels from such regions.

We formalize this intuition as follows. Suppose we would like to design a single query strategy $Q_1 : \mathcal{X} \to [0, 1]$ that determines the probability of querying the label for an instance during the active learning phase. For any $Q_1$, we have the following generalization error bound for learning with $n$ logged examples and $m$ unlabeled examples from which the learner can select and query for labels (for simplicity of illustration, we use the unclipped estimator here)

$$l(h_1) - l(h_2) \leq l(h_1, S) - l(h_2, S) + \frac{4 \log \frac{2|\mathcal{H}|}{\delta}}{3(m q_0 + n)} + \sqrt{4 \mathbb{E} \frac{\mathbb{1}\{h_1(X) \neq h_2(X)\}}{m Q_0(X) + n Q_1(X)} \log \frac{2|\mathcal{H}|}{\delta}}.$$

We propose to set $Q_1(x) = \mathbb{1}\{m Q_0(x) < \frac{m}{2} Q_0(x) + n\}$ which only queries instances if $Q_0(x)$ is small. This leads to fewer queries while guarantees an error bound close to the one achieved by setting $Q_1(x) \equiv 1$ that queries every instance. In Appendix E we give an example, Example 31, showing the reduction of queries due to this strategy.

The sample selection bias correction strategy is complementary to the DBAL technique. We note that a similar query strategy is proposed in [25], but the strategy here stems from a tighter analysis and can be applied with variance control techniques discussed in Section 4, and thus gives better label complexity guarantees as to be discussed in the analysis section.

### 5.1 Algorithm

Putting things together, our proposed algorithm is shown as Algorithm 1. It takes the logged data and an epoch schedule as input. It assumes the logging policy $Q_0$ and its distribution $f(x) = \Pr(Q_0(X) \leq x)$ are known (otherwise, these quantities can be estimated with unlabeled data).

Algorithm 1 uses the DBAL framework that recursively shrinks a candidate set $C$ and its corresponding disagreement region $D$ to save label queries by not querying examples outside $D$. In particular, at iteration $k$, it computes a clipping threshold $M_k$ (step 5) and MIS weights $w_k(x) := \frac{m + n_k}{m Q_0(X_i) + \sum_{j=1}^{k} \tau_i Q_i(X_i)}$ which are used to define the clipped MIS error estimator and two second moment estimators

$$l(h; \tilde{S}_k, M_k) := \frac{1}{m + n_k} \sum_{i=1}^{m+n_k} w_k(X_i) Z_i \mathbb{1}\{h(X_i) \neq \tilde{Y}_i\} \mathbb{1}\{w_k(X_i) \leq M_k\},$$

$$\hat{V}(h_1, h_2; \tilde{S}_k, M_k) := \frac{1}{m + n_k} \sum_{i=1}^{m+n_k} w_k^2(X_i) Z_i \mathbb{1}\{h_1(X_i) \neq h_2(X_i)\} \mathbb{1}\{w_k(X_i) \leq M_k\},$$

$$\hat{V}(h; \tilde{S}_k, M_k) := \frac{1}{m + n_k} \sum_{i=1}^{m+n_k} w_k^2(X_i) Z_i \mathbb{1}\{h(X_i) \neq \tilde{Y}_i\} \mathbb{1}\{w_k(X_i) \leq M_k\}.$$

The algorithm shrinks the candidate set $C_{k+1}$ by eliminating classifiers whose estimated error is larger than a threshold that takes the minimum empirical error and the second moment into account (step 7), and defines a corresponding disagreement region $D_{k+1} = \text{DIS}(C_{k+1})$ as the set of all instances on which there are two classifiers in the candidate set $C_{k+1}$ that predict labels differently. It derives a query policy $Q_{k+1}$ with the sample selection bias correction strategy (step 9). At the end of iteration $k$, it draws $\tau_{k+1}$ unlabeled examples. For each example $X$ with $Q_{k+1}(X) > 0$, if

$X \in D_{k+1}$, the algorithm queries for the actual label $Y$ and sets $\tilde{Y} = Y$, otherwise it infers the label and sets $\tilde{Y} = \hat{h}_k(X)$. These examples $\{X\}$ and their inferred or queried labels $\{\tilde{Y}\}$ are then used in subsequent iterations. In the last step of the algorithm, a classifier that minimizes the clipped MIS error with the second moment regularizer over all received data is returned.

---

**Algorithm 1** Disagreement-Based Active Learning with Logged Observational Data

---

1: Input: confidence $\delta$, logged data $T_0$, epoch schedule $\tau_1, \ldots, \tau_K$, $n = \sum_{i=1}^{K} \tau_i$.
2: $\tilde{S}_0 \leftarrow T_0$; $C_0 \leftarrow \mathcal{H}$; $D_0 \leftarrow \mathcal{X}$; $n_0 = 0$
3: **for** $k = 0, \ldots, K - 1$ **do**
4:      $\sigma_1(k, \delta, M) \leftarrow (\frac{M}{m+n_k} + \frac{M^2}{(m+n_k)^{\frac{3}{2}}}) \log \frac{|\mathcal{H}|}{\delta}$; $\sigma_2(k, \delta) = \frac{1}{m+n_k} \log \frac{|\mathcal{H}|}{\delta}$; $\delta_k \leftarrow \frac{\delta}{2(k+1)(k+2)}$
5:      Choose $M_k = \inf\{M \geq 1 \mid \frac{2M}{m+n_k} \log \frac{|\mathcal{H}|}{\delta_k} \geq \Pr(\frac{m+n_k}{mQ_0(X)+n_k} > M/2)\}$
6:      $\hat{h}_k \leftarrow \arg\min_{h \in C_k} l(h; \tilde{S}_k, M_k)$
7:      Define the candidate set $C_{k+1} \leftarrow \{h \in C_k \mid l(h; \tilde{S}_k, M_k) \leq l(\hat{h}_k; \tilde{S}_k, M_k) + \gamma_1 \sigma_1(k, \delta_k, M_k) + \gamma_1 \sqrt{\sigma_2(k, \delta_k)\hat{V}(h, \hat{h}_k; \tilde{S}_k, M_k)}\}$
8:      Define the Disagreement Region $D_{k+1} \leftarrow \{x \in \mathcal{X} \mid \exists h_1, h_2 \in C_{k+1} \text{ s.t. } h_1(x) \neq h_2(x)\}$
9:      $Q_{k+1}(x) \leftarrow \mathbb{1}\{mQ_0(x) + \sum_{i=1}^{k} \tau_i Q_i(x) < \frac{m}{2}Q_0(x) + n_{k+1}\}$;
10:     $n_{k+1} \leftarrow n_k + \tau_{k+1}$
11:     Draw $\tau_{k+1}$ samples $\{(X_t, Y_t)\}_{t=m+n_k+1}^{m+n_{k+1}}$, and present $\{X_t\}_{t=m+n_k+1}^{m+n_{k+1}}$ to the learner.
12:     **for** $t = m + n_k + 1$ to $m + n_{k+1}$ **do**
13:        $Z_t \leftarrow Q_{k+1}(X_t)$
14:        **if** $Z_t = 1$ **then**
15:           If $X_t \in D_{k+1}$, query for label: $\tilde{Y}_t \leftarrow Y_t$; otherwise infer $\tilde{Y}_t \leftarrow \hat{h}_k(X_t)$.
16:        **end if**
17:     **end for**
18:     $\tilde{T}_{k+1} \leftarrow \{X_t, \tilde{Y}_t, Z_t\}_{t=m+n_k+1}^{m+n_{k+1}}$, $\tilde{S}_{k+1} \leftarrow \tilde{S}_k \cup \tilde{T}_{k+1}$;
19: **end for**
20: Output $\hat{h} = \arg\min_{h \in C_K} l(h; \tilde{S}_K, M_k) + \gamma_1 \sqrt{\frac{1}{m+n} \log \frac{|\mathcal{H}|}{\delta_K} \hat{V}(h; \tilde{S}_K, M_k)}$.

---

## 5.2 Analysis

We have the following generalization error bound for Algorithm 1. Despite not querying for all labels, our algorithm achieves the same asymptotic bound as the one that queries labels for all online data.

**Theorem 4.** *Let $M = \inf\{M' \geq 1 \mid \frac{2M'}{m+n} \log \frac{|\mathcal{H}|}{\delta_K} \geq \Pr(\frac{m+n}{mQ_0(X)+n} \geq M'/2)\}$ be the final clipping threshold used in step 20. There is an absolute constant $c_0 > 1$ such that for any $\delta > 0$, with probability at least $1 - \delta$,*

$$l(\hat{h}) \leq l(h^\star) + c_0 \left( \sqrt{\mathbb{E}\frac{\mathbb{1}\{h^\star(X) \neq Y\}}{mQ_0(X)+n} \mathbb{1}\{\frac{m+n}{mQ_0(X)+n} \leq M\} \log \frac{|\mathcal{H}|}{\delta}} + \frac{M \log \frac{|\mathcal{H}|}{\delta}}{m+n} + \frac{M^2 \sqrt{\log \frac{|\mathcal{H}|}{\delta}}}{(m+n)^{\frac{3}{2}}} \right).$$

Next, we analyze the number of labels queried by Algorithm 1 with the help of following definitions.

**Definition 5.** *For any $t \geq 1, r > 0$, define the modified disagreement coefficient $\tilde{\theta}(r, t) := \frac{1}{r} \Pr\left( \text{DIS}(B(h^\star, r)) \cap \{x : Q_0(x) \leq \frac{1}{t}\} \right)$. Define $\tilde{\theta} := \sup_{r > 2\nu} \tilde{\theta}(r, \frac{2m}{n})$.*

The modified disagreement coefficient $\tilde{\theta}(r, t)$ measures the probability of the intersection of two sets: the disagreement region for the $r$-ball around $h^\star$ and where the propensity score $Q_0(x)$ is smaller than $\frac{1}{t}$. It characterizes the size of the querying region of Algorithm 1. Note that the standard disagreement coefficient [10], which is widely used for analyzing DBAL in the classical active learning setting, can be written as $\theta(r) := \tilde{\theta}(r, 1)$. Here, the modified disagreement coefficient modifies the standard definition to account for the reduction of the number of label queries due to the sample selection bias correction strategy: Algorithm 1 only queries examples on which $Q_0(x)$ is lower than some threshold, hence $\tilde{\theta}(r, t) \leq \theta(r)$. Moreover, our modified disagreement coefficient $\tilde{\theta}$ is always smaller than the modified disagreement coefficient of [25] (denoted by $\theta'$) which is used to analyze their algorithm.

Additionally, define $\alpha = \frac{m}{n}$ to be the size ratio of logged and online data, let $\tau_k = 2^k$, define $\xi = \min_{1 \leq k \leq K}\{M_k / \frac{m+n_k}{mq_0+n_k}\}$ to be the minimum ratio between the clipping threshold $M_k$ and maximum MIS weight $\frac{m+n_k}{mq_0+n_k}$ ($\xi \leq 1$ since $M_k \leq \frac{m+n_k}{mq_0+n_k}$ by the choice of $M_k$), and define $\bar{M} = \max_{1 \leq k \leq K} M_k$ to be the maximum clipping threshold. Recall $q_0 = \inf_X Q_0(X)$.

The following theorem upper-bounds the number of label queries by Algorithm 1.

**Theorem 6.** *There is an absolute constant $c_1 > 1$ such that for any $\delta > 0$, with probability at least $1 - \delta$, the number of labels queried by Algorithm 1 is at most:*

$$c_1 \tilde{\theta} \cdot (n\nu + \sqrt{\frac{n\nu\xi}{\alpha q_0 + 1} \log \frac{|\mathcal{H}| \log n}{\delta}} + \frac{\bar{M}\xi \log n}{\sqrt{n\alpha}} \sqrt{\log \frac{|\mathcal{H}| \log n}{\delta}} + \frac{\xi \log n}{\alpha q_0 + 1} \log \frac{|\mathcal{H}| \log n}{\delta}).$$

### 5.3 Discussion

In this subsection, we compare the theoretical performance of the proposed algorithm and some alternatives to understand the effect of proposed techniques. We present some empirical results in Section F in Appendix.

The theoretical performance of learning algorithms is captured by label complexity, which is defined as the number of label queries required during the active learning phase to guarantee the test error of the output classifier to be at most $\nu + \epsilon$ (here $\nu = l(h^\star)$ is the optimal error , and $\epsilon$ is the target excess error). This can be derived by combining the upper bounds on the error (Theorem 4) and the number of queries (Theorem 6).

- The label complexity is $\tilde{O}\left(\nu\tilde{\theta} \log |\mathcal{H}| \cdot \left(\frac{M}{\epsilon(1+\alpha)} + \frac{1}{\epsilon^2} \mathbb{E}\frac{\mathbb{1}\{h^\star(X) \neq Y\}}{1+\alpha Q_0(X)} \mathbb{1}\{\frac{1+\alpha}{1+\alpha Q_0(X)} \leq M\}\right)\right)$ for Algorithm 1. This is derived from Theorem 4, 6.

- The label complexity is $\tilde{O}\left(\nu\tilde{\theta} \log |\mathcal{H}| \cdot \left(\frac{1}{\epsilon(1+\alpha q_0)} + \frac{1}{\epsilon^2} \mathbb{E}\frac{\mathbb{1}\{h^\star(X) \neq Y\}}{1+\alpha Q_0(X)}\right)\right)$ without clipping. This is derived by setting the final clipping threshold $M_K = \frac{1+\alpha}{1+\alpha q_0}$. It is worse since $\frac{1+\alpha}{1+\alpha q_0} \geq M$.

- The label complexity is $\tilde{O}\left(\nu\tilde{\theta} \log |\mathcal{H}| \cdot (\frac{1}{\epsilon} + \frac{\nu}{\epsilon^2})\frac{1}{1+\alpha q_0}\right)$ if regularizers are removed further. This is worse since $\frac{\nu}{1+\alpha q_0} \geq \mathbb{E}\frac{\mathbb{1}\{h^\star(X) \neq Y\}}{1+\alpha Q_0(X)}$.

- The label complexity is $\tilde{O}\left(\nu\theta \log |\mathcal{H}| \cdot (\frac{1}{\epsilon} + \frac{\nu}{\epsilon^2})\frac{1}{1+\alpha q_0}\right)$ if we further remove the sample selection bias correction strategy. Here the standard disagreement coefficient $\theta$ is used ($\theta \geq \tilde{\theta}$).

- The label complexity is $\tilde{O}\left(\nu\theta \log |\mathcal{H}| \cdot \left(\frac{1}{\epsilon(1+\alpha q_0)} + \frac{\nu(q_0+\alpha)}{\epsilon^2(1+\alpha)^2 q_0}\right)\right)$ if we further remove the MIS technique. It can be shown $\frac{q_0+\alpha}{(1+\alpha)^2 q_0} \geq \frac{1}{1+\alpha q_0}$, so MIS gives a better label complexity bound.

- The label complexity is $\tilde{O}\left(\log |\mathcal{H}| \cdot \left(\frac{1}{\epsilon(1+\alpha q_0)} + \frac{\nu(q_0+\alpha)}{\epsilon^2(1+\alpha)^2 q_0}\right)\right)$ if DBAL is further removed. Here, all $n$ online examples are queried. This demonstrates that DBAL decreases the label complexity bound by a factor of $\nu\theta$ which is at most 1 by definition.

- Finally, the label complexity is $\tilde{O}\left(\nu\theta' \log |\mathcal{H}| \cdot \frac{\nu+\epsilon}{\epsilon^2}\frac{1}{1+\alpha q_0}\right)$ for [25], the only known algorithm in our setting. Here, $\theta' \geq \tilde{\theta}$, $\frac{\nu}{1+\alpha q_0} \geq \mathbb{E}\frac{\mathbb{1}\{h^\star(X) \neq Y\}}{1+\alpha Q_0(X)}$, and $\frac{1}{1+\alpha q_0} \geq \frac{M}{1+\alpha}$. Thus, the label complexity of the proposed algorithm is better than [25]. This improvement is made possible by the second moment regularizer, the principled clipping technique, and thereby the improved sample selection bias correction strategy.

## 6 Conclusion

We consider active learning with logged observational data where the learner is given an observational data set selected according to some logging policy, and can actively query for additional labels from an online data stream. Previous work applies disagreement-based active learning with an

importance weighted loss estimator to account for counterfactuals, which has high variance and leads to a high label complexity. In this work, we utilize variance control techniques for importance weighted estimators, and propose a novel variant of DBAL to make it amenable to variance-controlled importance sampling. Based on these improvements, a new sample selection bias correction strategy is proposed to further boost label efficiency. Our theoretical analysis shows that the proposed algorithm is statistically consistent and more label-efficient than prior work and alternative methods.

**Acknowledgement** We thank NSF under CCF 1513883 and 1719133 for support.

## Footnotes

[1]This generating process implies the standard unconfoundedness assumption in the counterfactual inference literature: $\Pr(Y_t, Z_t \mid X_t) = \Pr(Y_t \mid X_t) \Pr(Z_t \mid X_t)$. In other words, the label $Y_t$ is conditionally independent with the action $Z_t$ (indicating whether the label is observed) given the instance $X_t$.

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
