[Supplementary Material]

# A Preliminaries

## A.1 Summary of Key Notations

**Data** $T_0 = \{(X_t, Y_t, Z_t)\}_{t=1}^{m}$ is the logged data. $\tilde{T}_k = \{(X_t, \tilde{Y}_t, Z_t)\}_{t=m+n_{k-1}+1}^{m+n_k}$ ($1 \leq k \leq K$) is the online data collected in the $k$-th iteration of size $\tau_k = n_k - n_{k-1}$, and $\tilde{Y}_t$ equals either the actual label $Y_t$ drawn from the data distribution $D$ or the inferred label $\hat{h}_{k-1}(X_t)$ according to the candidate set $C_{k-1}$ at iteration $k-1$. $\tilde{S}_k = T_0 \cup \tilde{T}_1 \cup \cdots \cup \tilde{T}_k$.

For convenience, we additionally define $T_k = \{(X_t, Y_t, Z_t)\}_{t=m+n_{k-1}+1}^{m+n_k}$ to be the data set with the actual labels $Y_t$ drawn from the data distribution, and $S_k = T_0 \cup T_1 \cup \cdots \cup T_k$. The algorithm only observes $\tilde{S}_k$ and $\tilde{T}_k$, and $S_k, T_k$ are used for analysis only.

For $1 \leq k \leq K, n_k = \tau_1 + \cdots + \tau_k$, and we define $n_0 = 0$, $n = n_K$, $\tau_0 = m$. We assume $\tau_k \leq \tau_{k+1}$ for $1 \leq k < K$.

Recall that $\{(X_t, Y_t, Z_t)\}_{t=1}^{m+n}$ is an independent sequence, and furthermore $\{(X_t, Y_t)\}_{t=1}^{m+n}$ is an i.i.d. sequence drawn from $D$. For $(X, Z) \in T_k$ ($0 \leq k \leq K$), $Q_k(X) = \Pr(Z = 1 \mid X)$. Unless otherwise specified, all probabilities and expectations are over the random draw of all random variables $\{(X_t, Y_t, Z_t)\}_{t=1}^{m+n}$.

**Loss and Second Moment** The test error $l(h) = \Pr(h(X) \neq Y)$, the optimal classifier $h^\star = \arg\min_{h \in \mathcal{H}} l(h)$, and the optimal error $\nu = l(h^\star)$. At the $k$-th iteration, the Multiple Importance Sampling (MIS) weight $w_k(x) = \frac{m+n_k}{mQ_0(X_t) + \sum_{i=1}^{k} \tau_i Q_i(X_t)}$. The clipped MIS loss estimator $l(h; S_k, M) = \frac{1}{m+n_k} \sum_{i=1}^{m+n_k} w_k(X_i) Z_i \mathbb{1}\{h(X_i) \neq Y_i\} \mathbb{1}\{w_k(X_i) \leq M\}$. The (unclipped) MIS loss estimator $l(h; S_k) = l(h; S_k, \infty)$.

The clipped second moment $V(h; k, M) = \mathbb{E}[w_k(X) \mathbb{1}\{h(X) \neq Y\} \mathbb{1}\{w_k(X) \leq M\}]$, $V(h_1, h_2; k, M) = \mathbb{E}[w_k(X) \mathbb{1}\{h_1(X) \neq h_2(X)\} \mathbb{1}\{w_k(X) \leq M\}]$. The clipped second-moment estimators $\hat{V}(h; S_k, M) = \frac{1}{m+n_k} \sum_{i=1}^{m+n_k} w_k^2(X_i) Z_i \mathbb{1}\{h(X_i) \neq Y_i\} \mathbb{1}\{w_k(X_i) \leq M\}$, $\hat{V}(h_1, h_2; S_k, M) = \frac{1}{m+n_k} \sum_{i=1}^{m+n_k} w_k^2(X_i) Z_i \mathbb{1}\{h_1(X) \neq h_2(X)\} \mathbb{1}\{w_k(X_i) \leq M\}$. The unclipped second moments ($V(h; k), V(h_1, h_2; k)$) and second moment estimators ($\hat{V}(h; S_k), \hat{V}(h_1, h_2; S_k)$) are defined similarly.

**Disagreement Regions** The $r$-ball around $h$ is defined as $B(h, r) := \{h' \in \mathcal{H} \mid \Pr(h(X) \neq h'(X)) \leq r\}$, and the disagreement region of $C \subseteq \mathcal{H}$ is $\mathrm{DIS}(C) := \{x \in \mathcal{X} \mid \exists h_1 \neq h_2 \in C \text{ s.t. } h_1(x) \neq h_2(x)\}$.

The candidate set $C_k$ and its disagreement region $D_k$ are defined in Algorithm 1. The empirical risk minimizer (ERM) at $k$-th iteration $\hat{h}_k = \arg\min_{h \in C_k} l(h, \tilde{S}_k)$.

The modified disagreement coefficient $\tilde{\theta}(r, \alpha) := \frac{1}{r} \Pr\left(\mathrm{DIS}(B(h^\star, r)) \cap \{x : Q_0(x) \leq \frac{1}{\alpha}\}\right)$. $\tilde{\theta} = \sup_{r > 2\nu} \tilde{\theta}(r, \frac{2m}{n})$.

**Other Notations** $q_0 = \inf_x Q_0(x)$. $Q_{k+1}(X) = \mathbb{1}\{mQ_0(x) + \sum_{i=1}^{k} \tau_i Q_i(x) < \frac{m}{2}Q_0(x) + n_{k+1}\}$. $M_k = \inf\{M \geq 1 \mid \frac{2M}{m+n_k} \log \frac{|\mathcal{H}|}{\delta_k} \geq \Pr(\frac{m+n_k}{mQ_0(X)+n_k} > M/2)\}$. $\xi = \min_{1 \leq k \leq K}\{M_k / \frac{m+n_k}{mq_0+n_k}\}$. $\bar{M} = \max_{1 \leq k \leq K} M_k$.

## A.2 Elementary Facts

**Proposition 7.** *Suppose $a, c \geq 0$, $b \in \mathbb{R}$. If $a \leq b + \sqrt{ca}$, then $a \leq 2b + c$.*

*Proof.* Since $a \leq b + \sqrt{ca}$, $\sqrt{a} \leq \frac{\sqrt{c} + \sqrt{c+4b}}{2} \leq \sqrt{\frac{c+c+4b}{2}} = \sqrt{c + 2b}$ where the second inequality follows from the Root-Mean Square-Arithmetic Mean inequality. Thus, $a \leq 2b + c$. □

### A.3 Facts on Disagreement Regions and Candidate Sets

**Lemma 8.** *For any $k = 0, \ldots, K$, $M \geq 0$, if $h_1, h_2 \in C_k$, then $l(h_1; S_k, M) - l(h_2; S_k, M) = l(h_1; \tilde{S}_k, M) - l(h_2; \tilde{S}_k, M)$ and $\hat{V}(h_1, h_2; S_k, M) = \hat{V}(h_1, h_2; \tilde{S}_k, M)$.*

*Proof.* For any $(X_t, Y_t, Z_t) \in S_k$ that $Z_t = 1$, if $X_t \in \text{DIS}(C_k)$, then $Y_t = \tilde{Y}_t$, so $\mathbb{1}\{h_1(X_t) \neq Y_t\} - \mathbb{1}\{h_2(X_t) \neq Y_t\} = \mathbb{1}\{h_1(X_t) \neq \tilde{Y}_t\} - \mathbb{1}\{h_2(X_t) \neq \tilde{Y}_t\}$. If $X_t \notin \text{DIS}(C_k)$, then $h_1(X_t) = h_2(X_t)$, so $\mathbb{1}\{h_1(X_t) \neq Y_t\} - \mathbb{1}\{h_2(X_t) \neq Y_t\} = \mathbb{1}\{h_1(X_t) \neq \tilde{Y}_t\} - \mathbb{1}\{h_2(X_t) \neq \tilde{Y}_t\} = 0$. Thus, $l(h_1; S_k, M) - l(h_2; S_k, M) = l(h_1; \tilde{S}_k, M) - l(h_2; \tilde{S}_k, M)$.

$\hat{V}(h_1, h_2; S_k, M) = \hat{V}(h_1, h_2; \tilde{S}_k, M)$ holds since $\hat{V}(h_1, h_2; S_k, M)$ and $\hat{V}(h_1, h_2; \tilde{S}_k, M)$ do not involve labels $Y$ or $\tilde{Y}$. $\qquad\square$

The following lemmas are immediate from the definition.

**Lemma 9.** *For any $1 \leq k \leq K$, if $h \in C_k$, then $l(h; \tilde{S}_k, M) \leq l(h; S_k, M) \leq l(h; S_k)$, and $\hat{V}(h; \tilde{S}_k, M) \leq \hat{V}(h; S_k, M) \leq \hat{V}(h; S_k)$.*

*Remark* 10. The inequality on the second moment regularizer $\hat{V}$, which will be used to prove the error bound (Theorem 4) of Algorithm 1, is due to the decomposition property $\hat{V}(h; S_k, M) = \frac{|S_k \cap \text{DIS}(C_k)|}{m + n_k} \hat{V}(h; S_k \cap \text{DIS}(C_k), M) + \frac{|S_k \cap \text{DIS}(C_k)^c|}{m + n_k} \hat{V}(h; S_k \cap \text{DIS}(C_k)^c, M)$. It does not hold for estimated variance $\hat{\text{Var}}(h; S_k, M) := \hat{V}(h; S_k, M) - l(h; S_k, M)^2$. This explains the necessity of introducing the second moment regularizer.

**Lemma 11.** *For any $r \geq 2\nu$, any $\alpha \geq 1$, $\Pr(\text{DIS}(B(h^\star, r)) \cap \{x : Q_0(x) \leq \frac{1}{\alpha}\}) \leq r\tilde{\theta}(r, \alpha)$.*

### A.4 Facts on Multiple Importance Sampling Estimators

**Proposition 12.** *Let $f : \mathcal{X} \times \mathcal{Y} \to \mathbb{R}$. For any $k$, the following equations hold:*

$$\mathbb{E}[\frac{1}{m + n_k} \sum_{(X,Y,Z) \in S_k} w_k(X) Z f(X, Y)] = \mathbb{E}[f(X, Y)],$$

$$\mathbb{E}[\frac{1}{m + n_k} \sum_{(X,Y,Z) \in S_k} w_k^2(X) Z f(X, Y)] = \mathbb{E}[w_k(X) f(X, Y)].$$

*Proof.*

$$\mathbb{E}[\sum_{(X,Y,Z) \in S_k} w_k(X) Z f(X, Y)] = \sum_{i=0}^{k} \mathbb{E}[\sum_{(X,Y,Z) \in T_i} \mathbb{E}[w_k(X) f(X, Y) Z \mid X, Y]]$$

$$= \sum_{i=0}^{k} \mathbb{E}[\sum_{(X,Y,Z) \in T_i} w_k(X) f(X, Y) \mathbb{E}[Z \mid X, Y]]$$

$$\overset{(a)}{=} \sum_{i=0}^{k} \mathbb{E}[\sum_{(X,Y,Z) \in T_i} w_k(X) f(X, Y) \mathbb{E}[Z \mid X]]$$

$$= \sum_{i=0}^{k} \mathbb{E}[\sum_{(X,Y,Z) \in T_i} w_k(X) f(X, Y) Q_i(X)]$$

$$\overset{(b)}{=} \sum_{i=0}^{k} \tau_i \mathbb{E}[w_k(X) f(X, Y) Q_i(X)]$$

$$= \mathbb{E}[w_k(X) f(X, Y) \sum_{i=0}^{k} \tau_i Q_i(X)]$$

$$\overset{(c)}{=} (m + n_k) \mathbb{E}[f(X, Y)]$$

where (a) follows from $\mathbb{E}[Z \mid X] = \mathbb{E}[Z \mid X, Y]$ as $Z, Y$ are conditionally independent given $X$, (b) follows since $T_i$ is a sequence of i.i.d. random variables, and (c) follows from the definition $w_k(X) = \frac{m+n_k}{\sum_{i=0}^{k} \tau_i Q_i(X)}$.

The proof for the second equality is similar and skipped. $\qquad\square$

## A.5 Facts on the Sample Selection Bias Correction Query Strategy

The query strategy $Q_k$ can be simplified as follows.

**Proposition 13.** *For any* $1 \le k \le K$, $x \in \mathcal{X}$, $Q_k(x) = \mathbb{1}\{2n_k - mQ_0(x) > 0\}$.

*Proof.* The $k = 1$ case can be easily verified. Suppose it holds for $Q_k$, and we next show it holds for $Q_{k+1}$. Recall by definition $Q_{k+1}(x) = \mathbb{1}\{mQ_0(x) + \sum_{i=1}^{k} \tau_i Q_i(x) < \frac{m}{2} Q_0(x) + n_{k+1}\}$.

If $Q_k(x) = 1$, then $mQ_0(x) + \sum_{i=1}^{k-1} \tau_i Q_i(x) < \frac{m}{2} Q_0(x) + n_k$, so

$$mQ_0(x) + \sum_{i=1}^{k} \tau_i Q_i(x) < \frac{m}{2} Q_0(x) + n_k + \tau_k$$
$$\le \frac{m}{2} Q_0(x) + n_{k+1}$$

where the last inequality follows by the assumption on the epoch schedule $\tau_k \le \tau_{k+1} = n_{k+1} - n_k$. This implies $Q_{k+1}(x) = 1$. In this case, $\mathbb{1}\{2n_{k+1} - mQ_0(x) > 0\} = 1$ as well, since $n_{k+1} \ge n_k$ implies $2n_{k+1} - mQ_0(x) \ge 2n_k - mQ_0(x) > 0$.

The above argument also implies if $Q_k(x) = 0$, then $Q_1(x) = Q_2(x) = \cdots = Q_{k-1}(x) = 0$. Thus, if $Q_k(x) = 0$, then $Q_{k+1}(x) = \mathbb{1}\{mQ_0(x) < \frac{m}{2} Q_0(x) + n_{k+1}\} = \mathbb{1}\{2n_{k+1} - mQ_0(x) > 0\}$. $\quad\square$

The following proposition gives an upper bound of the multiple importance sampling weight, which will be used to bound the second moment of the loss estimators with the sample selection bias correction strategy.

**Proposition 14.** *For any* $1 \le k \le K$, $w_k(x) = \frac{m+n_k}{mQ_0(x) + \sum_{i=1}^{k} \tau_i Q_i(x)} \le \frac{m+n_k}{\frac{1}{2} mQ_0(x) + n_k}$.

*Proof.* The $k = 1$ case can be easily verified. Suppose it holds for $w_k$, and we next show it holds for $w_{k+1}$.

Now, if $Q_{k+1}(x) = 0$, then by Proposition 13, $2n_{k+1} - mQ_0(x) \le 0$, so $mQ_0(x) + \sum_{i=1}^{k+1} \tau_i Q_i(x) \ge mQ_0(x) \ge \frac{1}{2} mQ_0(x) + n_{k+1}$.

If $Q_{k+1}(x) = 1$, then by the induction hypothesis, $mQ_0(x) + \sum_{i=1}^{k+1} \tau_i Q_i(x) \ge \frac{1}{2} mQ_0(x) + n_k + \tau_{k+1} = \frac{1}{2} mQ_0(x) + n_{k+1}$.

Thus, in both cases, $mQ_0(x) + \sum_{i=1}^{k+1} \tau_i Q_i(x) \ge \frac{1}{2} mQ_0(x) + n_{k+1}$, so $w_{k+1}(x) \le \frac{m+n_{k+1}}{\frac{1}{2} mQ_0(x) + n_{k+1}}$. $\quad\square$

## A.6 Lower Bound Techniques

We present a lower bound for binomial distribution tails, which will be used to prove generalization error lower bounds.

**Lemma 15.** *Let* $0 < t < p < 1/2$, $B \sim Bin(n, p)$ *be a binomial random variable, and* $\delta = \sqrt{4n \frac{(t-p)^2}{p}}$. *Then,* $\Pr(B < nt) \ge \frac{1}{\sqrt{2\pi}} \frac{\delta}{\delta^2 + 1} \exp(-\frac{1}{2} \delta^2)$.

This Lemma is a consequence of following lemmas.

**Lemma 16.** *Suppose* $0 < p, q < 1$, $KL(p, q) = p \log \frac{p}{q} + (1-p) \log \frac{1-p}{1-q}$. *Then* $KL(p, q) \le \frac{(p-q)^2}{q(1-q)}$.

*Proof.* Since $\log x \le x - 1$, $p \log \frac{p}{q} + (1-p) \log \frac{1-p}{1-q} \le p(\frac{p}{q} - 1) + (1-p)(\frac{1-p}{1-q} - 1) = \frac{(p-q)^2}{q(1-q)}$. $\quad\square$

**Lemma 17.** *([5]) Suppose $X \sim N(0, 1)$, and define $\Phi(t) = \Pr(X \leq t)$. If $t > 0$, then $\Phi(-t) \geq \frac{1}{\sqrt{2\pi}} \frac{t}{t^2+1} \exp(-\frac{1}{2} t^2)$.*

**Lemma 18.** *([28]) Let $B \sim Bin(n, p)$ be a binomial random variable and $0 < k < np$. Then, $\Pr(B < k) \geq \Phi(-\sqrt{2nKL(\frac{k}{n}, p)})$.*

## B Deviation Bounds

In this section, we demonstrate deviation bounds for our error estimators on $S_k$.

We use following Bernstein-style concentration bound:

**Fact 19.** *Suppose $X_1, \ldots, X_n$ are independent random variables such that $|X_i| \leq M$. Then with probability at least $1 - \delta$,*

$$\left| \frac{1}{n} \sum_{i=1}^{n} X_i - \frac{1}{n} \sum_{i=1}^{n} \mathbb{E} X_i \right| \leq \frac{2M}{3n} \log \frac{2}{\delta} + \sqrt{\frac{2}{n^2} \sum_{i=1}^{n} \mathbb{E} X_i^2 \log \frac{2}{\delta}}.$$

**Theorem 20.** *For any $k = 0, \ldots, K$, any $\delta > 0$, if $\frac{2M \log \frac{|\mathcal{H}|}{\delta}}{m + n_k} \geq \Pr(\frac{m + n_k}{mQ_0(X) + n_k} \geq \frac{M}{2})$, then with probability at least $1 - \delta$, for all $h_1, h_2 \in \mathcal{H}$, the following statements hold simultaneously:*

$$|(l(h_1; S_k, M) - l(h_2; S_k, M)) - (l(h_1) - l(h_2))| \leq \frac{10 \log \frac{2|\mathcal{H}|}{\delta}}{3(m + n_k)} M + \sqrt{\frac{4 \log \frac{2|\mathcal{H}|}{\delta}}{m + n_k} V(h_1, h_2; k, M)};$$

(1)

$$|l(h_1; S_k, M) - l(h_1)| \leq \frac{10 \log \frac{2|\mathcal{H}|}{\delta}}{3(m + n_k)} M + \sqrt{\frac{4 \log \frac{2|\mathcal{H}|}{\delta}}{m + n_k} V(h_1; k, M)}.$$

(2)

*Proof.* We show proof for $k > 0$. The $k = 0$ case can be proved similarly.

First, define the clipped expected loss $l(h; k, M) = \mathbb{E}[\mathbb{1}\{h(X) \neq Y\} \mathbb{1}\{w_k(X) \leq M\}]$. We have

$$
\begin{aligned}
&|(l(h_1) - l(h_2)) - (l(h_1; k, M) - l(h_2; k, M))| \\
&= |\mathbb{E}[(\mathbb{1}\{h_1(X) \neq Y\} - \mathbb{1}\{h_2(X) \neq Y\}) \mathbb{1}\{w_k(X) > M\}]| \\
&\leq \mathbb{E}[\mathbb{1}[w_k(X) > M]] \\
&\leq \mathbb{E}[\mathbb{1}\{\frac{m + n_k}{mQ_0(X) + n_k} > \frac{M}{2}\}] \\
&\leq \frac{2M}{m + n_k} \log \frac{|\mathcal{H}|}{\delta}
\end{aligned}
$$

(3)

where the second inequality follows from Proposition 14, and the last inequality follows from the assumption on $M$.

Next, we bound $(l(h_1; S_k, M) - l(h_2; S_k, M)) - (l(h_1; k, M) - l(h_2; k, M))$.

For any fixed $h_1, h_2 \in \mathcal{H}$, define $N := |S_k|$, $U_t := w_k(X_t) Z_t \mathbb{1}\{w_k(X_t) \leq M\}(\mathbb{1}\{h_1(X_t) \neq Y_t\} - \mathbb{1}\{h_2(X_t) \neq Y_t\})$.

Now, $\{U_t\}_{t=1}^{N}$ is an independent sequence. $\frac{1}{N} \sum_{t=1}^{N} U_t = l(h_1; S_k, M) - l(h_2; S_k, M)$, and $\mathbb{E} \frac{1}{N} \sum_{t=1}^{N} U_t = l(h_1; k, M) - l(h_2; k, M)$ by Proposition 12. Moreover, since $(\mathbb{1}\{h_1(X_t) \neq Y_t\} - \mathbb{1}\{h_2(X_t) \neq Y_t\})^2 = \mathbb{1}\{h_1(X_t) \neq h_2(X_t)\}$, we have $\frac{1}{N} \sum_{t=1}^{N} U_t^2 = \hat{V}(h_1, h_2; S_k, M)$ and $\mathbb{E} \frac{1}{N} \sum_{t=1}^{N} U_t^2 = V(h_1, h_2; k, M)$ by Proposition 12. Applying Bernstein's inequality (Fact 19) to $\{U_t\}$, we have with probability at least $1 - \frac{\delta}{2}$,

$$\left| \frac{1}{N} \sum_{t=1}^{N} U_t - \mathbb{E} \frac{1}{N} \sum_{t=1}^{N} U_t \right| \leq \frac{2M}{3N} \log \frac{4}{\delta} + \sqrt{\frac{2}{N} V(h_1, h_2; k, M) \log \frac{4}{\delta}},$$

so $|(l(h_1; S_k, M) - l(h_2; S_k, M)) - (l(h_1; k, M) - l(h_2; k, M))| \leq \frac{2M}{3(m+n_k)} \log \frac{4}{\delta} + \sqrt{\frac{2}{m+n_k} V(h_1, h_2; k, M) \log \frac{4}{\delta}}$. By a union bound over $\mathcal{H}$, with probability at least $1 - \frac{\delta}{2}$ for all $h_1, h_2 \in \mathcal{H}$,

$$|(l(h_1; S_k, M) - l(h_2; S_k, M)) - (l(h_1; k, M) - l(h_2; k, M))|$$
$$\leq \frac{4M}{3(m+n_k)} \log \frac{2|\mathcal{H}|}{\delta} + \sqrt{\frac{4}{m+n_k} V(h_1, h_2; k, M) \log \frac{2|\mathcal{H}|}{\delta}}. \tag{4}$$

(1) follows by combining (3) and (4).

The proof for (2) is similar and skipped. $\qquad\square$

We use following bound for the second moment which is an immediate corollary of Lemmas B.1 and B.2 in [18]:

**Fact 21.** *Suppose* $X_1, \ldots, X_n$ *are independent random variables such that* $|X_i| \leq M$. *Then with probability at least* $1 - \delta$,

$$-\sqrt{\frac{2M^2}{n} \log \frac{1}{\delta}} - \frac{M^2}{n} \leq \sqrt{\frac{1}{n} \sum_{i=1}^{n} X_i^2} - \sqrt{\mathbb{E} \frac{1}{n} \sum_{i=1}^{n} X_i^2} \leq \sqrt{\frac{2M^2}{n} \log \frac{1}{\delta}}.$$

Recall by Lemma 12, $\mathbb{E}[\hat{V}(h_1, h_2; S_k, M)] = V(h_1, h_2; k, M)$ and $\mathbb{E}[\hat{V}(h_1; S_k, M)] = V(h_1; k, M)$. The following Corollary follows from the bound on the second moment.

**Corollary 22.** *For any* $k = 0, \ldots, K$, *any* $\delta, M > 0$, *with probability at least* $1 - \delta$, *for all* $h_1, h_2 \in \mathcal{H}$, *the following statements hold:*

$$\left| \sqrt{\hat{V}(h_1, h_2; S_k, M)} - \sqrt{V(h_1, h_2; k, M)} \right| \leq \sqrt{\frac{4M^2}{m+n_k} \log \frac{2|\mathcal{H}|}{\delta}} + \frac{M^2}{m+n_k}, \tag{5}$$

$$\left| \sqrt{\hat{V}(h_1; S_k, M)} - \sqrt{V(h_1; k, M)} \right| \leq \sqrt{\frac{4M^2}{m+n_k} \log \frac{2|\mathcal{H}|}{\delta}} + \frac{M^2}{m+n_k}. \tag{6}$$

**Corollary 23.** *There is an absolute constant* $\gamma_1$, *for any* $k = 0, \ldots, K$, *any* $\delta > 0$, *if* $\frac{2M \log \frac{|\mathcal{H}|}{\delta}}{m+n_k} \geq \Pr(\frac{m+n_k}{mQ_0(X)+n_k} \geq \frac{M}{2})$, *then with probability at least* $1 - \delta$, *for all* $h_1, h_2 \in \mathcal{H}$, *the following statements hold:*

$$|(l(h_1; S_k, M) - l(h_2; S_k, M)) - (l(h_1) - l(h_2))| \leq \gamma_1 \frac{M}{m+n_k} \log \frac{|\mathcal{H}|}{\delta} + \gamma_1 \frac{M^2}{(m+n_k)^{\frac{3}{2}}} \sqrt{\log \frac{|\mathcal{H}|}{\delta}} \tag{7}$$

$$+ \gamma_1 \sqrt{\frac{\log \frac{|\mathcal{H}|}{\delta}}{m+n_k} \hat{V}(h_1, h_2; S_k, M)};$$

$$l(h_1; S_k, M) \leq 2l(h_1) + \gamma_1 \frac{M}{m+n_k} \log \frac{|\mathcal{H}|}{\delta}. \tag{8}$$

*Proof.* Let event $E$ be the event that (1), (2), and (5) hold for all $h_1, h_2 \in \mathcal{H}$ with confidence $1 - \frac{\delta}{3}$ respectively. Assume $E$ happens (whose probability is at least $1 - \delta$).

(7) is immediate from (1) and (5).

For the proof of (8), apply (2) to $h_1$, we get

$$l(h_1; S_k, M) \leq l(h_1) + \frac{10 \log \frac{6|\mathcal{H}|}{\delta}}{3(m+n_k)} M + \sqrt{\frac{4 \log \frac{6|\mathcal{H}|}{\delta}}{m+n_k} \mathbf{V}(h_1; k, M)}.$$

Now, $\mathbf{V}(h_1; k, M) = \mathbb{E}[w_k(X)\mathbb{1}\{h_1(X) \neq Y\}\mathbb{1}\{w_k(X) \leq M\}] \leq M\mathbb{E}[\mathbb{1}\{h_1(X) \neq Y\}]$, so $\sqrt{\frac{4 \log \frac{6|\mathcal{H}|}{\delta}}{m+n_k} \mathbf{V}(h_1; k, M)} \leq \sqrt{\frac{4M \log \frac{6|\mathcal{H}|}{\delta}}{m+n_k} l(h_1)} \leq l(h_1) + \frac{M \log \frac{6|\mathcal{H}|}{\delta}}{(m+n_k)}$ where the last inequality follows from $\sqrt{ab} \leq \frac{a+b}{2}$ for $a, b \geq 0$, and (8) thus follows. $\square$

## C   Technical Lemmas for Disagreement-Based Active Learning

For any $0 \leq k < K$ and $\delta > 0$, define event $\mathcal{E}_{k,\delta}$ to be the event that the conclusions of Theorem 20 and Corollary 22 hold for $k$ with confidence $1 - \delta/2$ respectively. We have $\Pr(\mathcal{E}_{k,\delta}) \geq 1 - \delta$, and that $\mathcal{E}_{k,\delta}$ implies inequalities (7) and (8).

Recall that $\sigma_1(k, \delta, M) = \frac{M}{m+n_k} \log \frac{|\mathcal{H}|}{\delta} + \frac{M^2}{(m+n_k)^{\frac{3}{2}}} \sqrt{\log \frac{|\mathcal{H}|}{\delta}}$; $\sigma_2(k, \delta) = \frac{1}{m+n_k} \log \frac{|\mathcal{H}|}{\delta}$; $\delta_k = \frac{\delta}{2(k+1)(k+2)}$.

We first present a lemma which can be used to guarantee that $h^\star$ stays in candidate sets with high probability by induction.

**Lemma 24.** *For any $k = 0, \ldots K$, any $\delta > 0$, any $M \geq 1$ such that $\frac{2M \log \frac{|\mathcal{H}|}{\delta}}{m+n_k} \geq \Pr(\frac{m+n_k}{mQ_0(X)+n_k} \geq \frac{M}{2})$, on event $\mathcal{E}_{k,\delta}$, if $h^\star \in C_k$, then,*

$$l(h^\star; \tilde{S}_k, M) \leq l(\hat{h}_k; \tilde{S}_k, M) + \gamma_1 \sigma_1(k, \delta, M) + \gamma_1 \sqrt{\sigma_2(k, \delta) \hat{V}(h^\star, \hat{h}_k; \tilde{S}_k, M)}.$$

*Proof.*

$$l(h^\star; \tilde{S}_k, M) - l(\hat{h}_k; \tilde{S}_k, M)$$
$$= l(h^\star; S_k, M) - l(\hat{h}_k; S_k, M)$$
$$\leq \gamma_1 \sigma_1(k, \delta, M) + \gamma_1 \sqrt{\sigma_2(k, \delta) \hat{V}(h^\star, \hat{h}_k; S_k, M)}$$
$$= \gamma_1 \sigma_1(k, \delta, M) + \gamma_1 \sqrt{\sigma_2(k, \delta) \hat{V}(h^\star, \hat{h}_k; \tilde{S}_k, M)}$$

The first and the second equalities follow by Lemma 8. The inequality follows by Corollary 23. $\square$

Next, we present a lemma to bound the probability mass of the disagreement region of candidate sets.

**Lemma 25.** *Let $\hat{h}_{k,M} = \arg\min_{h \in C_k} l(h; \tilde{S}_k, M)$, and $C_{k+1}(\delta, M) := \{h \in C_k \mid l(h; \tilde{S}_k, M) \leq l(\hat{h}_{k,M}; \tilde{S}_k, M) + \gamma_1 \sigma_1(k, \delta, M) + \gamma_1 \sqrt{\sigma_2(k, \delta) \hat{V}(h, \hat{h}_{k,M}; \tilde{S}_k, M)}\}$. There is an absolute constant $\gamma_2 > 1$ such that for any $k = 0, \ldots, K$, any $\delta > 0$, any $M \geq 1$ such that $\frac{2M \log \frac{|\mathcal{H}|}{\delta}}{m+n_k} \geq \Pr(\frac{m+n_k}{mQ_0(X)+n_k} \geq \frac{M}{2})$, on event $\mathcal{E}_{k,\delta}$, if $h^\star \in C_k$, then for all $h \in C_{k+1}(\delta, M)$,*

$$l(h) - l(h^\star) \leq \gamma_2 \sigma_1(k, \delta, M) + \gamma_2 \sqrt{\sigma_2(k, \delta) M l(h^\star)}.$$

*Proof.* For any $h \in C_{k+1}(\delta, M)$, we have

$$l(h) - l(h^\star)$$

$$\leq l(h; S_k, M) - l(h^\star; S_k, M) + \frac{10M \log \frac{4|\mathcal{H}|}{\delta}}{3(m+n_k)} + \sqrt{4 \frac{\mathbf{V}(h^\star, h; k, M)}{m+n_k} \log \frac{4|\mathcal{H}|}{\delta}}$$

$$= l(h; \tilde{S}_k, M) - l(h^\star; \tilde{S}_k, M) + \frac{10M \log \frac{4|\mathcal{H}|}{\delta}}{3(m+n_k)} + \sqrt{4 \frac{\mathbf{V}(h^\star, h; k, M)}{m+n_k} \log \frac{4|\mathcal{H}|}{\delta}}$$

$$= l(h; \tilde{S}_k, M) - l(\hat{h}_{k,M}; \tilde{S}_k, M) + l(\hat{h}_{k,M}; \tilde{S}_k, M) - l(h^\star; \tilde{S}_k, M) + \frac{10M \log \frac{4\mathcal{H}}{\delta}}{3(m+n_k)} + \sqrt{4 \frac{\mathbf{V}(h^\star, h; k, M)}{m+n_k} \log \frac{4|\mathcal{H}|}{\delta}}$$

$$\leq \gamma_1 \sigma_1(k, \delta, M) + \gamma_1 \sqrt{\sigma_2(k, \delta) \hat{\mathbf{V}}(h, \hat{h}_{k,M}; \tilde{S}_k, M)} + \frac{10M \log \frac{4|\mathcal{H}|}{\delta}}{3(m+n_k)} + \sqrt{4 \frac{\mathbf{V}(h^\star, h; k, M)}{m+n_k} \log \frac{4|\mathcal{H}|}{\delta}}$$

$$(9)$$

where the first equality follows from Lemma 8, the first inequality follows from Theorem 20, and the second inequality follows from the definition of $C_k(\delta, M)$ and that $l(\hat{h}_{k,M}; \tilde{S}_k, M) \leq l(h^\star; \tilde{S}_k, M)$.

Next, we upper bound $\sqrt{\hat{\mathbf{V}}(h, \hat{h}_{k,M}; \tilde{S}_k, M)}$. We have

$$\sqrt{\hat{\mathbf{V}}(h, \hat{h}_{k,M}; \tilde{S}_k, M)} \leq \sqrt{\hat{\mathbf{V}}(h, h^\star; \tilde{S}_k, M) + \hat{\mathbf{V}}(h^\star, \hat{h}_{k,M}; \tilde{S}_k, M)}$$

$$\leq \sqrt{\hat{\mathbf{V}}(h, h^\star; \tilde{S}_k, M)} + \sqrt{\hat{\mathbf{V}}(h^\star, \hat{h}_{k,M}; \tilde{S}_k, M)}$$

where the first inequality follows from the triangle inequality that $\hat{\mathbf{V}}(h, \hat{h}_{k,M}; \tilde{S}_k, M) \leq \hat{\mathbf{V}}(h, h^\star; \tilde{S}_k, M) + \hat{\mathbf{V}}(h^\star, \hat{h}_{k,M}; \tilde{S}_k, M)$ and the second follows from the fact that $\sqrt{a+b} \leq \sqrt{a} + \sqrt{b}$ for $a, b \geq 0$.

For the first term, we have $\sqrt{\hat{\mathbf{V}}(h, h^\star; \tilde{S}_k, M)} = \sqrt{\hat{\mathbf{V}}(h, h^\star; S_k, M)} \leq \sqrt{\mathbf{V}(h, h^\star; k, M)} + \sqrt{\frac{4M^2}{m+n_k} \log \frac{4|\mathcal{H}|}{\delta}} + \frac{M^2}{m+n_k}$ by Corollary 22.

For the second term, we have

$$\sqrt{\hat{\mathbf{V}}(h^\star, \hat{h}_{k,M}; \tilde{S}, M)} \leq \sqrt{M(l(h^\star; \tilde{S}_k, M) + l(\hat{h}_{k,M}; \tilde{S}_k, M))}$$

$$\leq \sqrt{2Ml(h^\star; \tilde{S}_k, M)}$$

$$\leq \sqrt{2Ml(h^\star; S_k, M)}$$

$$\leq \sqrt{2M(2l(h^\star) + \gamma_1 \frac{M}{m+n_k} \log \frac{|\mathcal{H}|}{\delta})}$$

$$\leq \sqrt{\frac{2\gamma_1 M^2}{m+n_k} \log \frac{|\mathcal{H}|}{\delta}} + 2\sqrt{Ml(h^\star)}$$

where the first inequality follows since $w_k^2(X) Z \mathbb{1}\{h^\star(X) \neq \hat{h}_{k,M}(X)\} \mathbb{1}[w_k(X) \leq M] \leq M(w_k(X) Z \mathbb{1}\{h^\star(X) \neq Y\} + w_k(X) Z \mathbb{1}\{\hat{h}_{k,M}(X) \neq Y\})$, the second inequality follows since $l(\hat{h}_{k,M}; \tilde{S}_k, M) \leq l(h^\star; \tilde{S}_k, M)$, the third follows by Lemma 9 since we assume $h^\star \in C_k$, the fourth follows by Corollary 23, and the last follows by $\sqrt{a+b} \leq \sqrt{a} + \sqrt{b}$.

Therefore, $\sqrt{\hat{\mathbf{V}}(h, \hat{h}_{k,M}; \tilde{S}_k, M)} \leq \sqrt{\mathbf{V}(h, h^\star; k, M)} + (2 + \sqrt{2\gamma_1}) \sqrt{\frac{M^2}{m+n_k} \log \frac{4|\mathcal{H}|}{\delta}} + \frac{M^2}{m+n_k} + 2\sqrt{Ml(h^\star)}$. Continuing (9), we have

$$l(h) - l(h^\star) \le (\frac{10}{3} + 3\gamma_1 + 2\sqrt{2}\gamma_1^{\frac{3}{2}})\frac{M}{m + n_k} \log \frac{4|\mathcal{H}|}{\delta} + \gamma_1 \frac{M^2}{(m + n_k)^{\frac{3}{2}}} \sqrt{\log \frac{4|\mathcal{H}|}{\delta}}$$

$$+ (\gamma_1 + 2)\sqrt{\frac{\mathrm{V}(h^\star, h; k, M)}{m + n_k} \log \frac{4|\mathcal{H}|}{\delta}} + 2\gamma_1 \sqrt{\frac{Ml(h^\star)}{m + n_k} \log \frac{4|\mathcal{H}|}{\delta}}.$$

Now, since $w_k^2(X) Z \mathbb{1}\{h^\star(X) \neq \hat{h}_k(X)\} \mathbb{1}[w_k(X) \le M] \le M(w_k(X) Z \mathbb{1}\{h^\star(X) \neq Y\} + w_k(X) Z \mathbb{1}\{\hat{h}_k(X) \neq Y\})$, we have $\sqrt{\frac{\mathrm{V}(h^\star, h; k, M)}{m + n_k} \log \frac{4|\mathcal{H}|}{\delta}} \le \sqrt{\frac{M(l(h) - l(h^\star) + 2l(h^\star))}{m + n_k} \log \frac{4|\mathcal{H}|}{\delta}} \le \sqrt{\frac{M(l(h) - l(h^\star))}{m + n_k} \log \frac{4|\mathcal{H}|}{\delta}} + \sqrt{\frac{2Ml(h^\star)}{m + n_k} \log \frac{4|\mathcal{H}|}{\delta}}$ where the second follows by $\sqrt{a + b} \le \sqrt{a} + \sqrt{b}$ for $a, b \ge 0$.

Thus, $l(h) - l(h^\star) \le (\frac{10}{3} + 3\gamma_1 + 2\sqrt{2}\gamma_1^{\frac{3}{2}})\frac{M}{m+n_k} \log \frac{4|\mathcal{H}|}{\delta} + \gamma_1 \frac{M^2}{(m+n_k)^{\frac{3}{2}}}\sqrt{\log \frac{4|\mathcal{H}|}{\delta}} + (2\gamma_1 + \sqrt{2}\gamma_1 + 2\sqrt{2})\sqrt{\frac{Ml(h^\star)}{m+n_k} \log \frac{4|\mathcal{H}|}{\delta}} + (\gamma_1 + 2)\sqrt{\frac{M(l(h) - l(h^\star))}{m+n_k} \log \frac{4|\mathcal{H}|}{\delta}}.$

The result follows by applying Lemma 7 to $l(h) - l(h^\star)$. $\qquad\square$

## D  Proofs for Section 5.2

*Proof.* (of Theorem 4) Define event $\mathcal{E}^{(0)} := \bigcap_{k=0}^{K} \mathcal{E}_{k,\delta_k}$. By a union bound, $\Pr(\mathcal{E}^{(0)}) \ge 1 - \delta/2$. On event $\mathcal{E}^{(0)}$, by induction and Lemma 24, for all $k = 0, \ldots, K$, $h^\star \in C_k$.

$$l(\hat{h}) - l(h^\star) \le l(\hat{h}; S_K, M_K) - l(h^\star; S_K, M_K) + \gamma_1 \sigma_1(K, \delta_K, M_K) + \gamma_1 \sqrt{\sigma_2(K, \delta_K)\hat{\mathrm{V}}(\hat{h}, h^\star; S_K, M_K)}$$

$$= l(\hat{h}; \tilde{S}_K, M_K) - l(h^\star; \tilde{S}_K, M_K) + \gamma_1 \sigma_1(K, \delta_K, M_K) + \gamma_1 \sqrt{\sigma_2(K, \delta_K)\hat{\mathrm{V}}(\hat{h}, h^\star; \tilde{S}_K, M_K)}$$

$$\le l(\hat{h}; \tilde{S}_K, M_K) + \gamma_1 \sqrt{\sigma_2(K, \delta_K)\hat{\mathrm{V}}(\hat{h}; \tilde{S}_K, M_K)} - l(h^\star; \tilde{S}_K, M_K) - \gamma_1 \sqrt{\sigma_2(K, \delta_K)\hat{\mathrm{V}}(h^\star; \tilde{S}_K, M_K)}$$

$$\quad + \gamma_1 \sigma_1(K, \delta_K, M_K) + 2\gamma_1 \sqrt{\sigma_2(K, \delta_K)\hat{\mathrm{V}}(h^\star; \tilde{S}_K, M_K)}$$

$$\le \gamma_1 \sigma_1(K, \delta_K, M_K) + 2\gamma_1 \sqrt{\sigma_2(K, \delta_K)\hat{\mathrm{V}}(h^\star; \tilde{S}_K, M_K)}$$

$$\le \gamma_1 \sigma_1(K, \delta_K, M_K) + 2\gamma_1 \sqrt{\sigma_2(K, \delta_K)\hat{\mathrm{V}}(h^\star; S_K, M_K)}$$

$$\le 3\gamma_1 \sigma_1(K, \delta_K, M_K) + 2\gamma_1 \sqrt{\sigma_2(K, \delta_K)\mathrm{V}(h^\star; K, M_K)}$$

where the equality follows from Lemma 8, the first inequality follows from Corollary 23, the second follows as $\sqrt{\hat{\mathrm{V}}(\hat{h}, h^\star; \tilde{S}_K, M_K)} \le \sqrt{\hat{\mathrm{V}}(\hat{h}; \tilde{S}_K, M_K) + \hat{\mathrm{V}}(h^\star; \tilde{S}_K, M_K)} \le \sqrt{\hat{\mathrm{V}}(\hat{h}; \tilde{S}_K, M_K)} + \sqrt{\hat{\mathrm{V}}(h^\star; \tilde{S}_K, M_K)}$, the third follows from the definition of $\hat{h}$, the forth follows from Lemma 9, and the last follows from Corollary 22. $\qquad\square$

*Proof.* (of Theorem 6) Define event $\mathcal{E}^{(0)} := \bigcap_{k=0}^{K} \mathcal{E}_{k,\delta_k}$. On this event, by induction and Lemma 24, for all $k = 0, \ldots, K - 1$, $h^\star \in C_k$, and consequently by Lemma 25, $D_{k+1} \subseteq \mathrm{DIS}(B(h^\star, 2\nu + \epsilon_k))$ where $\epsilon_k = \gamma_2 \sigma_1(k, \delta_k, M_k) + \gamma_2 \sqrt{\sigma_2(k, \delta_k)M_k \nu}$.

For any $k = 0, \ldots K - 1$, the number of label queries at iteration $k$ is $U_k := \sum_{t=m+n_k+1}^{m+n_{k+1}} Z_t \mathbb{1}\{X_t \in D_{k+1}\}$ where the RHS is a sum of i.i.d. Bernoulli random variables with expectation $\mathbb{E}[Z_t \mathbb{1}\{X_t \in D_{k+1}\}] = \Pr(D_{k+1} \cap \{x : Q_0(x) < \frac{2n_{k+1}}{m}\})$ since $Z_t = Q_{k+1}(x) = \mathbb{1}\{2n_{k+1} - mQ_0(x) > 0\}$ by Proposition 13. A Bernstein inequality implies that on an event $\mathcal{E}^{(1,k)}$ of probability at least $1 - \delta_k/2$, $U_k \le 2\tau_{k+1} \Pr(D_{k+1} \cap \{x : Q_0(x) < \frac{2n_{k+1}}{m}\}) + 2\log \frac{4}{\delta_k}$.

Define $\mathcal{E}^{(1)} := \bigcap_{k=0}^{K-1} \mathcal{E}^{(1,k)}$, and $\mathcal{E}^{(2)} := \mathcal{E}^{(0)} \cap \mathcal{E}^{(1)}$. By a union bound, we have $\Pr(\mathcal{E}^{(2)}) \geq 1 - \delta$. Now, on event $\mathcal{E}^{(2)}$, for any $k < K$, $D_{k+1} \subseteq \mathrm{DIS}(B(h^\star, 2\nu + \epsilon_k))$, so by Lemma 11 $\Pr(D_{k+1} \cap \{x : Q_0(x) < \frac{2n_{k+1}}{m}\}) \leq (2\nu + \epsilon_k)\tilde{\theta}(2\nu + \epsilon_k, \frac{2n_{k+1}}{m})$. Therefore, the total number of label queries

$$
\begin{aligned}
\sum_{k=0}^{K-1} U_k \leq & \tau_1 + \sum_{k=1}^{K-1} 2\tau_{k+1} \Pr(D_{k+1} \cap \{x : Q_0(x) < \frac{2n_{k+1}}{m}\}) + 2K \log \frac{4}{\delta_K} \\
\leq & 1 + 2 \sum_{k=1}^{K-1} \tau_{k+1}(2\nu + \epsilon_k)\tilde{\theta}(2\nu + \epsilon_k, \frac{2n_{k+1}}{m}) + 2K \log \frac{4}{\delta_K} \\
\leq & 1 + 2K \log \frac{4}{\delta_K} + 2\tilde{\theta}(2\nu + \epsilon_{K-1}, \frac{2n}{m}) \cdot \bigg( 2n\nu \\
& + \gamma_2 \sum_{k=1}^{K-1} \Big( \frac{\tau_{k+1} M_k}{m + n_k} \log \frac{|\mathcal{H}|}{\delta_k} + \frac{\tau_{k+1} M_k^2}{(m + n_k)^{\frac{3}{2}}} \sqrt{\log \frac{|\mathcal{H}|}{\delta_k}} + \tau_{k+1} \sqrt{\frac{M_k}{m + n_k} \nu \log \frac{|\mathcal{H}|}{\delta_k}} \Big) \bigg).
\end{aligned}
$$

Recall that $\alpha = \frac{m}{n}, \tau_k = 2^k$, $\xi = \min_{1 \leq k \leq K}\{M_k / \frac{m + n_k}{mq_0 + n_k}\}$, $\bar{M} = \max_{1 \leq k \leq K} M_k$. We have $\sum_{k=1}^{K-1} \frac{\tau_{k+1} M_k}{m + n_k} \leq \sum_{k=1}^{K-1} \frac{\xi \tau_k}{mq_0 + n_k} \leq \sum_{k=1}^{K} \frac{\xi n_k}{\alpha n_k q_0 + n_k} \leq \frac{K\xi}{\alpha q_0 + 1}$ where the first inequality follows as $\frac{M_k}{m + n_k} \leq \frac{\xi}{mq_0 + n_k}$, and the second follows by $m = n\alpha \geq n_k \alpha$. Besides, $\sum_{k=1}^{K-1} \frac{\tau_k M_k^2}{(m + n_k)^{\frac{3}{2}}} \leq \sum_{k=1}^{K-1} \frac{\tau_k M_k \xi}{\sqrt{m + n_k}(mq_0 + n_k)} \leq \sum_{k=1}^{K-1} \frac{\bar{M}\xi}{\sqrt{m + n_k}} \leq \frac{K\bar{M}\xi}{\sqrt{n\alpha}}$ where the first inequality follows as $\frac{M_k}{m + n_k} \leq \frac{\xi}{mq_0 + n_k}$, and the second follows as $M_k \leq \bar{M}$ and $\tau_k \leq mq_0 + n_k$. Finally, $\sum_{k=1}^{K} \tau_k \sqrt{\frac{M_k}{m + n_k}} \leq \sum_{k=1}^{K} \sqrt{\frac{\tau_k \xi}{\alpha q_0 + 1}} \leq \sqrt{\frac{n\xi}{\alpha q_0 + 1}}$ where the first inequality follows as $\frac{M_k}{m + n_k} \leq \frac{\xi}{mq_0 + n_k}$ and $mq_0 + n_k \geq \tau_k(\alpha q_0 + 1)$.

Therefore,

$$
\begin{aligned}
\sum_{k=0}^{K-1} U_k \leq & 1 + 2K \log \frac{4}{\delta_K} + 2\tilde{\theta}(2\nu + \epsilon_{K-1}, \frac{2n}{m}) \bigg( 2n\nu \\
& + \gamma_2 \Big( \frac{K\xi}{\alpha q_0 + 1} \log \frac{K^2|\mathcal{H}|}{\delta} + \frac{K\bar{M}\xi}{\sqrt{n\alpha}} \sqrt{\log \frac{K^2|\mathcal{H}|}{\delta}} + \sqrt{\frac{n\xi\nu}{\alpha q_0 + 1} \log \frac{K^2|\mathcal{H}|}{\delta}} \Big) \bigg).
\end{aligned}
$$

$\square$

# E   Proofs and Examples for Sections 4 and 5

**Generalization Error Bound**

Theorem 1 and Corollary 3 are immediate from the following theorem.

**Theorem 26.** *Let* $\hat{h}_M = \arg\min_{h \in \mathcal{H}} l(h; S, M) + \sqrt{\frac{\lambda}{m} \hat{V}(h; S, M)}$. *For any* $\delta > 0$, $M \geq 1$, $\lambda \geq 4 \log \frac{|\mathcal{H}|}{\delta}$, *with probability at least* $1 - \delta$ *over the choice of* $S$,

$$
\begin{aligned}
l(\hat{h}_M) - l(h^\star) \leq & \frac{2\lambda M}{m} + \frac{16M}{3m} \log \frac{|\mathcal{H}|}{\delta} + \frac{M^2}{m^{\frac{3}{2}}} \sqrt{4 \log \frac{|\mathcal{H}|}{\delta}} \\
& + \sqrt{\frac{\lambda}{m} \mathbb{E} \frac{\mathbb{1}\{h^\star(X) \neq Y\}}{Q_0(X)} \mathbb{1}[\frac{1}{Q_0(X)} \leq M]} + \Pr_X(\frac{1}{Q_0(X)} > M).
\end{aligned}
\tag{10}
$$

*Proof.* The proof is similar to the proofs for Theorem 4 and 20, and is omitted. $\square$

## Second Moment Regularizer

*Proof.* (of Theorem 2) For any $0 < \nu < \frac{1}{3}$, $m > \frac{49}{\nu^2}$, set $q_0 = \frac{1}{40}\nu$, $c = \frac{1}{3}$, $\epsilon = \frac{c^2 + \sqrt{c^4 + 4c^2 q_0 \nu m}}{2 q_0 m}$. It can be checked that $\epsilon < \nu$ and $m = c^2 \frac{\nu + \epsilon}{q_0 \epsilon^2}$. Let $\mathcal{X} = \{x_1, x_2, x_3\}$, and define $\Pr(X = x_1) = \nu$, $\Pr(X = x_2) = \nu + \epsilon$, $\Pr(X = x_3) = 1 - 2\nu - \epsilon$, and $\Pr(Y = 1) = 1$. Let $\mathcal{H} = \{h_1, h_2\}$ where $h_1(x_1) = -1$, $h_1(x_2) = h_1(x_3) = 1$, and $h_2(x_2) = -1$, $h_2(x_1) = h_2(x_3) = 1$. Define the logging policy $Q_0(x_1) = Q_0(x_3) = 1$, $Q_0(x_2) = q_0$. Let $S = \{(X_t, Y_t, Z_t)\}_{t=1}^m$ be a dataset of size $m$ generated from the aforementioned distribution. Clearly, we have $l(h_1) = \nu$ and $l(h_2) = \nu + \epsilon$. We next prove that $\Pr(l(h_1, S) > l(h_2, S)) \geq \frac{1}{100}$. This implies that with probability at least $\frac{1}{100}$, $h_2$ is the minimizer of the importance weighted loss $l(h, S)$, and its population error $\Pr(h_2(X) \neq Y) = \nu + \epsilon = \nu + \frac{1}{q_0 m} + \sqrt{\frac{\nu}{q_0 m}}$.

We have

$$\Pr(l(h_1, S) > l(h_2, S)) \geq \Pr(l(h_1, S) > \nu - \frac{\epsilon}{2} \text{ and } l(h_2, S) < \nu - \frac{\epsilon}{2})$$

$$= 1 - \Pr(l(h_1, S) \leq \nu - \frac{\epsilon}{2} \text{ or } l(h_2, S) \geq \nu - \frac{\epsilon}{2})$$

$$\geq 1 - \Pr(l(h_1, S) \leq \nu - \frac{\epsilon}{2}) - \Pr(l(h_2, S) \geq \nu - \frac{\epsilon}{2})$$

$$= \Pr(l(h_2, S) < \nu - \frac{\epsilon}{2}) - \Pr(l(h_1, S) \leq \nu - \frac{\epsilon}{2})$$

Observe that by our construction, $m l(h_1, S) = \sum_{i=1}^m \mathbb{1}\{X_i = x_1\}$ follows the binomial distribution $\text{Bin}(m, \nu)$. By a Chernoff bound, $\Pr(l(h_1, S) \leq \nu - \frac{\epsilon}{2}) \leq e^{-\frac{1}{2}m\epsilon^2}$. Since $\epsilon \geq \sqrt{\frac{c^2 \nu}{q_0 m}} \geq \sqrt{\frac{40 c^2}{m}}$, $e^{-\frac{1}{2}m\epsilon^2} \leq e^{-20c^2} = e^{-\frac{20}{9}}$.

By our construction, we also have that $q_0 m l(h_2, S) = \sum_{i=1}^m \mathbb{1}\{X_i = x_2, Z_i = 1\}$ which follows the binomial distribution $\text{Bin}(m, q_0(\nu + \epsilon))$. Thus, $\Pr(l(h_2, S) \leq \nu - \frac{\epsilon}{2}) = \Pr(q_0 m l(h_2, S) \leq q_0 m(\nu + \epsilon) - \frac{3}{2}q_0 m\epsilon) \geq \frac{1}{\sqrt{2\pi}} \frac{3c}{9c^2 + 1} e^{-\frac{9}{2}c^2} = \frac{1}{2\sqrt{2\pi}} e^{-\frac{1}{2}}$ where the inequality follows by Lemma 15.

Therefore, $\Pr(l(h_1, S) > l(h_2, S)) \geq \Pr(l(h_2, S) < \nu - \frac{\epsilon}{2}) - \Pr(l(h_1, S) \leq \nu - \frac{\epsilon}{2}) \geq \frac{1}{2\sqrt{2\pi}} e^{-\frac{1}{2}} - e^{-\frac{20}{9}} \geq \frac{1}{100}$. $\square$

*Remark* 27. A similar result for general cost-sensitive empirical risk minimization is proved in [17, 18]. In [17, 18], they construct examples where $\text{Var}(h^\star) = 0$ and learning $h^\star$ with unregularized ERM gives $\tilde{\Omega}(\sqrt{\frac{1}{m}})$ error, while regularized ERM gives $\tilde{O}(\frac{1}{m})$ error. However, their construction does not work in our setting because the bound for unregularized ERM [25] also gives $\tilde{O}(\frac{1}{m})$ error when $\text{Var}(h^\star) = 0$ (since $\text{Var}(h^\star) = 0$ implies $l(h^\star) = 0$), so more careful construction and analysis are needed.

## Clipping

The clipping threshold $M_0$ is chosen to minimize an error bound for the clipped second-moment regularized ERM. According to Theorem 26, we would like to choose $M$ that minimizes the RHS of (10). We set $\lambda = 4 \log \frac{|\mathcal{H}|}{\delta}$ in Theorem 26, focus on the low order terms with respect to $m$, and minimize

$$e(M) := \sqrt{\frac{4 \log \frac{|\mathcal{H}|}{\delta}}{m} \mathbb{E} \frac{1}{Q_0(X)} \mathbb{1}[\frac{1}{Q_0(X)} \leq M]} + \Pr_X(\frac{1}{Q_0(X)} > M) \text{ instead since } \mathbb{1}\{h^\star(X) \neq Y\}$$

could not be determined with unlabeled samples. In this sense, the following proposition shows that our choice of $M$ is nearly optimal.

**Proposition 28.** *Suppose random variable $\frac{1}{Q_0(X)}$ has a probability density function, and there exists $M_0 \geq 1$ such that $\frac{2 \log \frac{|\mathcal{H}|}{\delta}}{m} M_0 = \Pr_X(\frac{1}{Q_0(X)} > M_0)$. Then $e(M_0) \leq \sqrt{2} \inf_{M \geq 1} e(M)$.*

*Proof.* Define $f_1(M) = \frac{4 \log \frac{|\mathcal{H}|}{\delta}}{m} \mathbb{E} \frac{1}{Q_0(X)} \mathbb{1}[\frac{1}{Q_0(X)} \leq M]$, and $f_2(M) = \Pr_X(\frac{1}{Q_0(X)} > c)$. We first show that $f_1(M_0) + f_2(M_0)^2 \leq \inf_{M > 1} f_1(M) + f_2(M)^2$.

Let $g(x)$ be the probability density function of random variable $1/Q_0(X)$. We have $f_1(M) = \frac{4\log\frac{|\mathcal{H}|}{\delta}}{m}\int_0^M xg(x)\,dx$ and $f_2(M) = \int_M^\infty g(x)\,dx$, so $f_1'(M) = \frac{4\log\frac{|\mathcal{H}|}{\delta}}{m}Mg(M)$, and $f_2'(M) = -g(M)$. Define $f(M) = f_1(M) + f_2(M)^2$. We have

$$f'(M) = f_1'(M) + 2f_2'(M)f_2(M)$$
$$= 2g(M)(\frac{2\log\frac{|\mathcal{H}|}{\delta}}{m}M - f_2(M)).$$

Recall we assume there exists $M_0 \geq 1$ such that $\frac{2\log\frac{|\mathcal{H}|}{\delta}}{m}M_0 = f_2(M_0)$. Since $\frac{2\log\frac{|\mathcal{H}|}{\delta}}{m}M$ is strictly increasing w.r.t. $M$ and $f_2(M)$ is non-increasing w.r.t. $M$, it follows that $f(M)$ achieves its minimum at $M_0$, that is, for any $c \geq 1$, $f_1(M_0) + f_2^2(M_0) \leq f_1(M) + f_2^2(M)$.

Now, $\sqrt{f_1(M_0) + f_2^2(M_0)} \geq \frac{1}{\sqrt{2}}(\sqrt{f_1(M_0)} + f_2(M_0))$ since $\sqrt{a+b} \geq \frac{1}{\sqrt{2}}(\sqrt{a} + \sqrt{b})$ for any $a, b \geq 0$, and $\sqrt{f_1(M) + f_2^2(M)} \leq \sqrt{f_1(M)} + f_2(M)$ since $\sqrt{a+b} \leq \sqrt{a} + \sqrt{b}$ for any $a, b \geq 0$. Thus $\frac{1}{\sqrt{2}}(\sqrt{f_1(M_0)} + f_2(M_0)) \leq \sqrt{f_1(M)} + f_2(M)$ for all $M > 0$, which concludes the proof. $\square$

*Remark* 29. Since $\frac{1}{M}\Pr_X(\frac{1}{Q_0(X)} > M)$ is monotonically decreasing with respect to $M$ and its range is $(0, 1)$, the existence and uniqueness of $M_0$ are guaranteed if $\frac{2}{m}\log\frac{|\mathcal{H}|}{\delta} < 1$.

The following example shows that our choice of $M$ indeed avoids outputting suboptimal classifiers.

**Example 30.** Let $\mathcal{X} = \{x_0, x_1, x_2, x_3, x_4\}$, $\mathcal{H} = \{h_1, h_2, h_3, h_4\}$. Suppose $\Pr(Y = 1) = -1$, $\nu < \frac{1}{10}$, $\alpha < 0.01$, and $\epsilon = \frac{\nu}{1+1/100\alpha}$. The marginal distribution on $X$, the prediction of each classifier, and the logging policy $Q_0$ is defined in Table 1.

Table 1: An example for clipping

|  | $x_0$ | $x_1$ | $x_2$ | $x_3$ | $x_4$ |
|---|---|---|---|---|---|
| $h_1(\cdot)$ | 1 | 1 | -1 | -1 | -1 |
| $h_2(\cdot)$ | 1 | -1 | 1 | -1 | -1 |
| $h_3(\cdot)$ | 1 | -1 | -1 | 1 | -1 |
| $h_4(\cdot)$ | -1 | -1 | -1 | -1 | 1 |
| $\Pr_X(\cdot)$ | $\nu - \epsilon$ | $\epsilon$ | $4\epsilon$ | $16\epsilon$ | $1 - \nu - 20\epsilon$ |
| $Q_0(\cdot)$ | 1 | $\alpha$ | $\alpha$ | $4\alpha$ | $4\alpha$ |

We have $l(h_1) = \nu$, $l(h_2) = \nu + 3\epsilon$, $l(h_3) = \nu + 15\epsilon$, $l(h_4) = 1 - \nu - 20\epsilon$. Next, we consider when examples with $Q_0$ equals $\alpha$, i.e. examples on $x_1$ and $x_2$, should be clipped. We set the failure probability $\delta = 0.01$.

If $m \geq \frac{28}{\alpha\epsilon}$, without clipping our error bound guarantees that (by minimizing a regularized training error) learner can achieve an error of less than $\nu + 3\epsilon$, so it would output the optimal classifier $h_1$ with high probability. On the other hand, if $M < \frac{1}{\alpha}$, then all examples on $x_1$ and $x_2$ are ignored due to clipping, so the learner would not be able to distinguish between $h_1$ and $h_2$, and thus with constant probability the error of the output classifier is at least $l(h_2) = \nu + 3\epsilon$. This means if $m \geq \frac{28}{\alpha\epsilon}$, examples on $x_1$ and $x_2$ should not be clipped.

If $m \geq \frac{2}{\alpha\epsilon}$ and examples on $x_1$ and $x_2$ are clipped, our error bound guarantees learner can achieve an error of less than $\nu + 16\epsilon$, which means the learner would output either $h_1$ or $h_2$ and achieve an actual error of at most $\nu + 3\epsilon$. However, without clipping, the learner would require $m \geq \frac{4}{\alpha\epsilon}$ to achieve an error of less than $\nu + 16\epsilon$. Thus, if $m \leq \frac{4}{\alpha\epsilon}$, examples on $x_1$ and $x_2$ should be clipped.

To sum up, examples with $Q_0$ equals $\alpha$ (i.e. $x_1$ and $x_2$) should be clipped if $m \leq \frac{4}{\alpha\epsilon}$ and not be clipped if $m \geq \frac{28}{\alpha\epsilon}$. Our choice of the clipping threshold clips $x_1$ and $x_2$ whenever $m \leq \frac{24}{5\alpha\epsilon}$, which falls inside the desired interval.

**Sample Selection Bias Correction Strategy**

The following example shows the sample selection bias correction strategy indeed improves label complexity.

**Example 31.** Let $\lambda > 1$ be any constant. Suppose $\mathcal{X} = \{x_1, x_2\}$, $Q_0(x_1) = 1$, $Q_0(x_2) = \alpha$, $\Pr(x_1) = 1 - \mu$, $\Pr(x_2) = \mu$ and assume $\mu \leq \frac{1}{4\lambda}$ and $\alpha \leq \frac{\mu^2}{2\lambda}$. Assume the logged data size $m$ is greater than twice as the online stream size $n$. Without the sample selection bias correction strategy, after seeing $n$ examples, the learner queries all $n$ examples and achieves an error bound of $\frac{4 \log \frac{2|\mathcal{H}|}{\delta}}{3(m\alpha+n)} + \sqrt{4(\frac{c\mu}{m+n} + \frac{\mu}{m\alpha+n}) \log \frac{2|\mathcal{H}|}{\delta}}$ by minimizing the regularized MIS loss. With the sample selection bias correction strategy, the learner only queries $x_2$, so after seeing $n$ examples, it queries only $\mu n$ examples in expectation and achieves an error bound of $\frac{4 \log \frac{2|\mathcal{H}|}{\delta}}{3(m\alpha+n)} + \sqrt{4(\frac{c\mu}{m} + \frac{\mu}{m\alpha+n}) \log \frac{2|\mathcal{H}|}{\delta}}$. With some algebra, it can be shown that to achieve the same error bound, if $\frac{\lambda\alpha}{\mu}m \leq n \leq \frac{\mu}{2}m$, then the number of queries requested by the learner without the sample selection bias correction correction strategy is at least $\lambda$ times more than the number of queries for the learner with the bias correction strategy. Since this holds for any $\lambda \geq 1$, the decrease of the number of label queries due to our sample selection bias correction strategy can be significant.

# F    Experiments

We conduct experiments to compare the performance of the proposed active learning algorithm against some baseline methods. Our experiment results confirm our theoretical analysis that the test error of the proposed algorithm drops faster than alternative methods as the number of label queries increases.

## F.1    Methodology

**Algorithms and Implementations**

We consider the following algorithms:

- PASSIVE: A passive learning algorithm that queries labels for all examples. It directly optimizes an importance weighted estimator;

- ACTIVE18: The active learning algorithm proposed in [25]. It applies the disagreement-based active learning framework, multiple importance sampling, and a sample selection bias correction strategy.

- ACTIVEVC: Algorithm 1 proposed in this paper. It applies the *variance controlled* disagreement-based active learning framework, multiple importance sampling, and an *improved* sample selection bias correction strategy.

Similar to [25], our implementation of disagreement-based active learning framework follows the Vowpal Wabbit ([1]) package. In particular,

- We set the hypothesis space to be the set of linear classifiers, and replace the 0-1 loss with a squared loss.

- We do not explicitly maintain the candidate set $C_k$ or the disagreement region $D_k$. To compute $\hat{h}_k$ in line 6 of Algorithm 1, we ignore the constraint $h \in C_k$ and conduct online gradient descent with step size $\sqrt{\frac{\eta}{\eta+t}}$. To approximately check whether $x \in D_{k+1}$ in line 15, let $w_k$ be the normal vector for $\hat{h}_k$, and $a$ be current step size. We claim $x \in D_{k+1}$ if $\frac{|2w_k^\top x|}{a x^\top x} \leq \sqrt{\frac{C \cdot \hat{V}(\hat{h}_k; \tilde{S}_k, M_k)}{m+n_k}} + \frac{C \cdot M_k}{m+n_k}$. Here $C$ is a parameter that captures the model capacity (this corresponds to the $\log \frac{|\mathcal{H}|}{\delta}$ term in the error bound; as noted in [12], this is often loose and needs to be tuned as a parameter in practice) and we tune this parameter in experiments.

Besides, we incorporate variance-controlled importance sampling into active learning through the following way:

| (a) Certainty | (b) Uncertainty |

Figure 1: Test error vs the number of labels under different logging policies with the best parameters.

- In order to find the clipping threshold $M_k$ (line 5 of Algorithm 1), we empirically estimate $\Pr(\frac{m+n_k}{mQ_0(X)+n_k} > M/2)$ on the logged observational data (note that this estimation does not involve labels).
- We follow [22] to approximately calculate the online gradient for optimization with a variance regularizer.

**Data**

We generate a synthetic dataset where 6000 examples are drawn uniformly at random from $[0, 1]^{30}$, and labels are assigned by a linear separator and get flipped with probability 0.05. We randomly split the dataset into 80% training data and 20% test data. Among the training dataset, we randomly choose around 50% as logged observational data, and apply a synthetic logging policy to choose which labels in the observational data set are revealed to the algorithm. Our experiments use the following two policies:

- Certainty: We first find a linear hyperplane that approximately separates the data. Then, we reveal the label with a higher probability (i.e., larger $Q_0$ value) if the example is further away from this hyperplane.
- Uncertainty: We first find a linear hyperplane that approximately separates the data. Then, we reveal the label with a higher probability (i.e., larger $Q_0$ value) if the example is closer to this hyperplane.

**Parameter Tuning**

We follow [13] and [25] to tune the model capacity $C$ and learning rate $\eta$, and report the best result for each algorithm under each logging policy.

In particular, let $e(i, A, p, l)$ be the test error of algorithm $A$ with parameter set $p = (C, \eta)$ after making $l$ label queries during the $i$-th trial ($i = 1, 2, \ldots, N$). We evaluate the performance of the algorithm $A$ with parameter set $p$ by following Area Under the error-label Curve metric: $\text{AUC}(A, p) = \frac{1}{2N} \sum_{i=1}^{N} \sum_{l} (e(i, A, p, l+1) + e(i, A, p, l))$. At the end, for each algorithm, we report the error-label curve achieved with the parameter set $p$ that minimizes $\text{AUC}(A, p)$.

In our experiments, we try $C$ in $\{0.01 \times 2^i \mid i = 0, 2, 4, \cdots, 10\}$, and $\eta$ in $\{0.0001 \times 2^i \mid i = 0, 2, 4, \cdots, 12\}$. For each algorithm, policy, and parameter set, the experiments are repeated for $N = 16$ times.

**F.2    Results**

We plot test error as a function of the number of labels in Figure 1. It shows that test errors achieved by the proposed method drop faster than both the passive learning baseline, and the prior work [25]

which does not apply variance control techniques. Additionally, as the number of labels grows, the gap widens.