[Reviews · NeurIPS 2019]

Reviewer 1



It is difficult for me to judge how important is the addressed problem from the practical viewpoint. However, it definitely seems interesting from the theoretical perspective, as it involves non-trivial interactions between influence-weighted empirical risk minimization (IW-ERM) and the DBAL framework, which motivates the authors to modify these two approaches in a number of non-trivial ways in their framework. These modifications, as summarized below, may be of independent interest to the semi-supervised and online learning communities. -First, the authors suggest to replace regularization by the estimated variance of the empirical risk in IW-ERM by the estimated second moment regularization. This trick allows to incorporate the additionally queried observations due to the separability of the empirical second moment in the sample. On the other hand, given a sufficiently large sample size, the estimated second moment regularization has similar performance (in absence of additionally queried data) to the estimated variance one. -The second adjustment to IW-ERM is clipping of the loss according to the value of the probability $Q_0(X)$ with which they are preserved by the logging policy; specifically, one only preserves the data points with large enough value of $Q_0(X)$ in the empirical risk objective. Here the authors propose a principled way of choosing the clipping threshold, and provide a generalization bound in absence of the active learning stage. -Then a few adjustments to the DBAL framework of Hanneke et al. are proposed, allowing to: (1) incorporate the observational sample $T_0$ directly into the DBAL learning process, instead of merely using it for warm-start; this is achieved by the novel technique called Multiple Importance Sampling (MIS), and provably results in an improved label complexity estimate; (2) preserve the statistical consistency of DBAL in the presence of empirical variance/second moment regularization, by modifying the definition of the DBAL candidate set; (3) correct for the sample selection bias in the observational data, in the active learning stage, by making the querying probability $Q_1(x)$ dependent on the logging probability $Q_0(X)$. This idea was previously suggested in the work of Yan et al. ([22]), but the current version of the technique is shown to lead to better label complexity. Overall I think it is solid piece of theoretical work that deserves to be accepted at NeurIPS. The authors succeeded in demonstrating that the combination of counterfactual and active learning requires original ideas. Their framework results in the efficient algorithm that admits better theoretical guarantees than the state of the art (these improvements are thoroughly discussed), while at the same time remaining practically applicable. The main issue I see with the paper is the absence of experiments to complement the theoretical results. In particular, it would be interesting to evaluate the proposed ideas in numerical experiments with real or simulated data, in order to compare the relative (practical) impact of the proposed techniques (cf. the discussion in Sec. 5.3). More broadly, it would also be interesting to see how much better the whole framework performs compared to the classical DBAL with warm-start using the observational data. Besides, I have some minor remarks: 1. It is unclear how much clipping improves the performance of importance sampling (cf. Thms 3-4 vs. Thm 1); it is only shown that it cannot degrade. It would be useful to have a quantitative bound showing that clipping can provably *increase* the statistical and label complexities under some assumptions. 2. While the estimated second moment regularizer decomposes over the sample, which results in a simpler analysis, the resulting generalization bound should generally be weaker than the one for the estimated variance regularization. This is not entirely clear from the current text (cf. lines 140 to 146), and I encourage the authors to comment on the rates. In this connection, it would also be interesting to consider alternative regularization / finer analysis that combines the benefits of the two approaches. 3. I would encourage the authors to explain Algorithm 1 (Sec. 5.1) in more detail, perhaps by compressing the introductory sections, or putting comparison with related work into Appendix. 4. Overall the paper is well-written; a few typos: correctionstrategy'' and correctiontechnique'' throughout the text; L90: a instance space'' L140: a error rates'' L145: the second square-root expression seems to be incorrect L185: other alternative methods''

Reviewer 2



The paper is on the active version of a learning model where the learner receives a logged'' data set, meaning that the label of each example x is observed with a given logging probability $Q_0(x)$ (known to the learner). In the active version there is also an additional data set, where the learner can decide whether to request the label of an example. The paper proves various generalization error bounds in this model. This model has been studied in the recent paper [22]. The approach of the paper is closely related to the approach of [22], and modifies the learning algorithm in several respects. Section 5.3 gives a detailed quantitative comparison of the label complexities of several versions of the algorithm, and the algorithm of [22]. The label complexities depend on several parameters involved and therefore the comparisons are not straightforward. This is a solid technical paper. The main problem seems to be that no effort is made to justify the relevance and significance of the results, both in terms of the model and of the significance of the improvements over previous work. The model is a recent one, and therefore explanation of its relevance would be necessary. For example, the assumption that the learner knows the logging policy seems to require some explanation. Also, the results are only meaningful if $q_0$ is not too small, and that could also deserve some discussion. For active learning, the usual assumption is that there are many unlabeled examples. Here it is mentioned at the end of Section 3 that the number of unlabeled examples is at most the size of logged data set. No justification or comment is provided as to why this assumption is reasonable, or even why it is technically necessary. Comment on response: it would be useful to add the comment on the effect of not knowing the logging policy, possibly with some explanation of how its approximate knowledge enters in the bounds.

Reviewer 3



This is a well-written and quite interesting paper achieving best-known theoretical results in a well-motivated learning setting. The authors consider a setting in which a subpopulation, sampled from the general population according to a known policy, is labeled, and the goal is to produce a reliable classifier for the general population (counterfactual learning). The algorithm additionally has access to a stream of unlabeled examples from the general population, which it can actively query for labels. The main difficulty in the counterfactual setting is overcoming the high variance that results when the subsampling policy undersamples some features. But directly addressing this variance using standard techniques works poorly in the online, active setting because the minimizer of regularized risk changes with each additional sample, making it challenging to preserve consistency guarantees. The authors address these challenges using a basket of techniques that are more than the sum of their parts. Instead of regularizing based on the variance, they use the second moment directly for their regularizer. Instead of maintaining the best hypothesis with respect to regularized loss within their candidate set, they ensure that the best hypothesis with respect to prediction error remains in the candidate set. Instead of querying solely based on disagreement within the candidate set, they first sample according to a policy designed to reduce bias resulting from the observed subpopulation. Individually, each modification to existing approaches is fairly minor, but they combine to an algorithm with provably good sample complexity. Furthermore, the authors show how each of their algorithmic improvements is necessary for their sample complexity bounds. Altogether, this paper represents an excellent foundation for an area deserving of further study. ===== After reviewing author feedback, I have revised my score slightly downward. I hope this paper is accepted, but I also hope the authors will make a serious attempt to provide intuition for the myriad parameters and bounds they introduce.

[Author Response · NeurIPS 2019]

We thank all the reviewers for their feedback. We will incorporate them in the future version.

Overall, reviewers agree that our paper is technically sound. Both Reviewers 1 & 3 acknowledge that our setup is novel
and well-motivated, and our techniques and results are non-trivial and significant. R2 has some questions about our
setting and comparison against previous work [22], which we can clarify.

**Problem Setting (R2)**.

Our problem setting lies at the intersection of active learning and counterfactual inference, and has connections to
learning under sample-selection bias and covariate shift. All of these very well-studied problems in machine learning
that are far from solved. The novelty in our setting lies in using active learning to reduce the need for labels in
counterfactual inference.

"the learner knows the logging policy": This is a fairly standard assumption in prior work [1,5,19,22] and holds for
example in online advertisement. In many applications where it does not hold, one can estimate the logging policy by
fitting a model (for example, Athey et al, "Approximate residual balancing: debiased inference of average treatment
effects in high dimensions." and references therein). Note that this fitting can be done from unlabeled data only.

"the number of unlabeled examples is at most the size of logged data set": We look at this setting mostly for simplicity,
and our algorithm will work (with minor modifications) with more unlabeled data so long as the labeling budget in the
online phase is limited. Our algorithm is most useful in a setting where the learner has already collected some logged
data and would like to query a few more labels to build a classifier for the population. Allowing unlimited data and
labeling budget in the online phase will lead to trivial solutions that do not use the logged data.

**Comparison with Prior Work (R2)**

**Novelty and significance:** We apply and analyze two variance control techniques (regularization and clipping), and
demonstrate how to combine them with the disagreement-based active learning (DBAL) framework in order to derive a
better sample selection bias correction method. Note that combining DBAL with regularization is technically non-trivial,
and the outcome (regularized DBAL) may be of independent interest.

**The role of $\tilde{\theta}$:** Our results are in line with a long line of prior work on active learning theory. Prior work shows that
changing the interaction mode or setup in active learning typically leads to a constant factor improvement in label
complexity – which is measured by a modified form of the disagreement coefficient – see for example Zhang and
Chaudhuri. "Active learning from weak and strong labelers." NeurIPS 2015; Huang, et al. "Active learning with oracle
epiphany." NeurIPS 2016.

**Value of $q_0$** ("the results are only meaningful if $q_0$ is not too small"): Quite contrary, because we do active learning, our
method does work well even if $q_0$ is small (unlike passive learning solutions). In particular, our label complexity depends
on an average term $\mathbb{E}[\frac{1}{1+\alpha Q_0(X)}]$, while the label complexity of [22] and many other baselines is either proportional to
$1/q_0$ or $\frac{1}{1+\alpha q_0}$ which can be much worse.

**Experiments (R1 and R2)**

The main contribution of this paper is a new algorithm with theoretical analysis. We agree that it would be interesting to
see how the proposed algorithm works practically, and we will add some experiments to the final version.

**Discussion of Results (R1, R2, R3)**

Gain from the clipping technique: The exact gain from the clipping technique depends on the data distribution. We
provide a concrete example (Example 30) in our paper. We will make it clearer and provide more quantitative analysis
in the final version.

Difference between error bounds for the variance regularizer and the second moment regularizer: The error bound is
about $\tilde{O}(\sqrt{\frac{1}{m}\mathbb{E}\frac{1[h^*(X)\neq Y]}{Q_0(X)}})$ with the second moment regularizer while $\tilde{O}(\sqrt{\frac{1}{m}\mathrm{Var}(\frac{1[h^*(X)\neq Y]Z}{Q_0(X)})})$ with the variance
regularizer. The latter is smaller, but the difference is almost negligible since $\frac{1}{m}\mathbb{E}\frac{1[h^*(X)\neq Y]}{Q_0(X)} - \frac{1}{m}\mathrm{Var}(\frac{1[h^*(X)\neq Y]Z}{Q_0(X)}) =$
$\frac{1}{m}l(h^*)^2$ diminishes as $m \to \infty$.

Parameters in Section 5.3: Thanks for pointing this out. We have provided some examples in Appendix (proof of
Theorem 2, Examples 30, 31) , and we will elaborate them and make it clearer in the final version. In terms of R3's
question about the inequality at line 298, the difference between its LHS and RHS depends can be quite significant for
some data distribution and logging policy. For example, if $Q_0(X) = q_0$ with a very small probability and is close to 1
elsewhere, then LHS is still about $\frac{\nu}{1+\alpha q_0}$ while RHS is about $\frac{\nu}{1+\alpha}$ which is much smaller if $\alpha$ is large and $q_0$ is small.

[Meta-Review · NeurIPS 2019]

The paper is a refinement of a previous work of from ICML 2018. The bounds are sometimes significantly better. The main reservation expressed by reviewers is a lack of quantitative comparison to that prior work [22], and generally some understanding of whether this new work finally identifies the "right" dependence on these various quantities. Some interesting examples could help with the first issue, and proving lower bounds could help with the second issue. (It follows from known results that \tilde{\theta} would not show up in a lower bound, but it would still be good to know whether the new bound has the right dependence on the other quantities, say in the case that \tilde{\theta} is O(1)). I believe the paper is acceptable. However, I would like to strongly suggest that the authors try to address these issues in the camera ready version, ideally providing interesting (and non-contrived) examples to illustrate the improvements in the new bound, and trying to provide a lower bound that reflects what the "right" dependence on various parameters is.